# Estimation of single-cell and tissue perturbation effect in spatial transcriptomics via Spatial Causal Disentanglement

**Stathis Megas**[*]
University of Cambridge, Cambridge Center for AI in Medicine, Wellcome Sanger Institute
`em613@cam.ac.uk`

**Daniel G. Chen**[*] **& Krzysztof Polanski**
University of Cambridge, Wellcome Sanger Institute

**Moshe Eliasof & Carola-Bibiane Schönlieb**
University of Cambridge, Department of Applied Mathematics and Theoretical Physics

**Sarah A. Teichmann**
University of Cambridge, Wellcome Sanger Institute
`sat1003@cam.ac.uk`

## Abstract

Models of Virtual Cells and Virtual Tissues at single-cell resolution would allow us to test perturbations in silico and accelerate progress in tissue and cell engineering. However, most such models are not rooted in causal inference and as a result, could mistake correlation for causation. We introduce Celcomen, a novel generative graph neural network grounded in mathematical causality to disentangle intra- and inter-cellular gene regulation in spatial transcriptomics and single-cell data. Celcomen can also be prompted by perturbations to generate spatial counterfactuals, thus offering insights into experimentally inaccessible states, with potential applications in human health. We validate the model's disentanglement and identifiability through simulations, and demonstrate its counterfactual predictions in clinically relevant settings, including human glioblastoma and fetal spleen, recovering inflammation-related gene programs post immune system perturbation. Moreover, it supports mechanistic interpretability, as its parameters can be reverse-engineered from observed behavior, making it an accessible model for understanding both neural networks and complex biological systems.

## 1 Introduction

A cell's gene expression profile simultaneously encodes information about its intrinsic characteristics and extrinsic tissue microenvironment. Disentangling these two effects and understanding the causal links between them are necessary to fully reconstruct the complex interplay of intra- and inter-cellular interactions in human tissues during homeostasis and post-disease or therapy-induced perturbation (Rood et al., 2024; Megas et al., 2024b; Bunne et al., 2024). To achieve this, a robust framework for causal disentanglement is needed.

Causal inference seeks to uncover the mechanisms that generate the observed data by leveraging the mathematical principle of identifiability (Pearl, 2009; Khemakhem et al., 2020; Xi & Bloem-Reddy, 2023; Zhang et al., 2023b). This principle holds when there exists a single unique model that fits the data, and thus we are assured that our observations can only be explained by the given model. However, despite the several useful properties that directly follow from identifiability (e.g.

---

[*]Contributed equally to this work.

robustness, generalizability, and self-consistency), the vast majority of current deep learning models violate this principle (Khemakhem et al., 2020; Somepalli et al., 2022). Moreover, applications of causal inference in spatial context have been limited (Reich et al., 2021; Papadogeorgou & Samanta, 2024) and there is a need for causally identifiable methods in spatial causal inference.

One of the goals in representation learning includes disentangling the data into conceptually distinct variables to achieve a representation of it that is more interpretable, generalizable, and suitable for reasoning (Squires et al., 2023). The simplest setting of conceptual distinctness of variables is that of statistical independence, which can be approached using independent component analysis (Comon, 1994). However, data and our reasoning about it often involves variables that are not statistically independent but are linked together in a causal acyclic graph (Pearl, 2009). For example, the expression of a ligand gene $\alpha$ in a cell $i$ is not independent of but *causes* the expression of a target gene $\beta$ in a neighboring cell $j$.

The generalization of learning statistically independent variables to learning sets of variables that are linked together in a causal graph is referred to as causal disentanglement (Squires et al., 2023), and combines the insights of disentangled representation learning and causal inference. In its most ambitious form, causal disentanglement aims to learn both the latent variables and the causality structure of the underlying causality diagram with identifiability guarantees (Zhang et al., 2023b). Although disentangled representations offer more interpretable descriptions of the data and insights into the inner workings of neural networks, the emerging field of mechanistic interpretability (Bereska & Gavves, 2024; Ferrando & Voita, 2024; Rai et al., 2024) has the even bigger ambition to completely specify a neural network's computation. In its most granular form, this entails comprehensively reverse-engineering the model's weights, which is of particular interest to safety alignment of LLMs (Bereska & Gavves, 2024).

Concurrently with the theoretical developments in causal inference and machine learning, technology developments, such as VisiumHD (Nagendran et al., 2023), Curio, Stereo-seq (Cheng et al., 2023) and Xenium (Salas et al., 2023), have allowed profiling the gene expression of cells at single-cell resolution as well as the cell's spatial coordinates in the tissue. These datasets, referred to as spatial transcriptomics, have also enabled us to perform perturbation experiments, such as gene knock-outs, in spatial samples and at large scale (Binan et al., 2023). This advancement has created the need for joint causal modeling of cell and tissue architecture (Rood et al., 2024; Megas et al., 2024b), to better capture the causal links of gene regulation.

In this paper, we introduce Celcomen to address the problem of inferring and disentangling the causal structure diagram of feature interactions from spatial samples, as illustrated in Fig. 1A. A typical example is gene regulation within cells compared with the behavior across nearest neighbors. That is, we aim to understand the causal links between features within a node, but also the causal links across nodes, see Fig. 1B, which might be confounded by spurious correlations see Fig. 1C,D. We then use it to perform spatial perturbations, such as spatial gene knock-outs, in selected regions of a tissue to help guide efforts of cell and tissue engineering. In other words, we here aim to introduce a model of Virtual Tissues. Models of Virtual Cells predict the effect that changes in the micro- and macro-environment of the cell (such as perturbing the age of the donor, the tissue the cell is in, the drug treatment, knock-outs from guide RNAs etc) have on gene expression (Bunne et al., 2024; Roohani et al., 2024; Megas et al., 2024b). Conversely, a model of Virtual Tissues aims to not only estimate the effect the environment has on the cell but also the effect that the cell has on its environment and overall tissue.

**Main Contributions.** The contributions of this paper are two-fold, and offer advancements from both (i) computational biology and (ii) machine learning aspects. In terms of (i), the contributions include:

- A proof of concept that generalizing models of Virtual Cells to models of Virtual Tissues is possible.
- A novel framework of causal structure learning for feature interaction in graph data, such as gene-gene interactions in spatial transcriptomics data.
- Gene regulation inference by integrating *both* dissociated and spatial single-cell data.
- A framework to perform counterfactuals on spatial transcriptomics graphs, such as answering the question "What would the cells in this tissue have looked like had we performed a gene knock-out in a specific location in the tissue?".

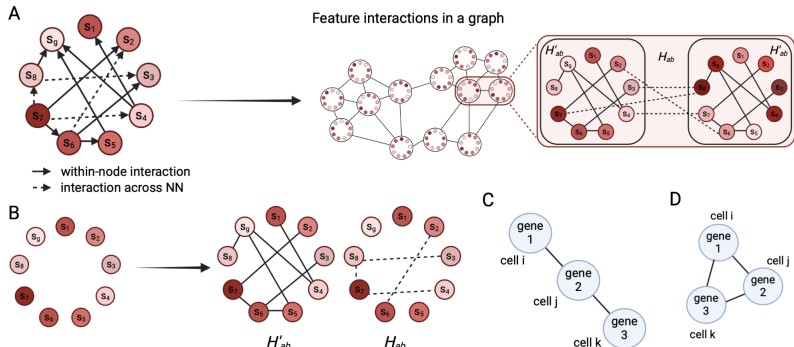

Figure 1: A) Structural causal model (SCM) between feature variables $s_1, ..., s_g$. The directed acyclic graph (DAG) of the SCM has edges of different types, depending on whether a variable causally influences another in the same node, or the nearest neighbor nodes (or nodes further away). This causality diagram generates spatial data such as spatial transcriptomics data. B) Our method is able to 1) disentangle the causality graph into its two components, and 2) retrieve the undirected Markov equivalence class of the graph of each component. Our model infers and disentangles the multi-type SCM based on samples of those features organized in space. C) Gene 1 causes gene 2 to be expressed in the neighboring cell and gene 2 causes gene 3 in its neighbor. D) Half of the time the 1-neighbor of a 1-neighbor is also a 1-neighbor leading to spurious colocalization of genes 1 and 3 although there is no causal link.

- To our knowledge, the first causally identifiable model for spatial transcriptomics analysis.

In terms of (ii), this paper offers:

- The construction of a k-hop GNN whose network parameters are interpretable, and guaranteed to assume the values of the problem's Lagrange multipliers, which are meaningful, physical quantities.
- A toy model for mechanistic interpretability, since Celcomen's weights can be recovered from its input-output computation.
- We provide a theoretical understanding of the model, namely, we mathematically prove that the problem of spatial causal disentanglement is identifiable. We also computationally verify our theoretical findings.

## 2 RELATED WORK

**Cell-communication.** Several previous works on cell communication have relied on prior knowledge of protein-protein interactions (PPI) or gene regulatory networks (GRN) to distinguish intrinsic and extrinsic circuits of gene regulation; this reliance often excludes key cell-cell interaction partners that are unreported (Browaeys et al., 2020; da Rocha et al., 2022). Recent deep learning models advance on this limitation by simultaneously modeling intrinsic and extrinsic features; however, these models lack interpretable insight due to their black-box nature (Schaar et al., 2024). Further, virtually all current models lack mathematical (identifiability) guarantees (Khemakhem et al., 2020), leading to their hyper-sensitivity to input data variability; exceedingly few accept both spatial and single-cell input data (Birk et al., 2024; Bernstein et al., 2022); and many cannot perform in silico perturbation experiments critical to understanding tissue behavior during disease (Cang et al., 2023; Wilk et al., 2024; Jerby-Arnon & Regev, 2022). While these works have introduced marked computational leaps in spatial transcriptomics, they often cannot perform causal inferences due to their lack of identifiability which mathematically prevents many current models from deriving comprehensive mechanistic insights into cell and tissue biology. Altough applications of causal inference and counterfactual predictions in spatial transcriptomic have been limited, there are by now several such methods in the dissociated single-cell field achieving state of the art performance (Tejada-Lapuerta et al., 2023; Aliee et al., 2023; Piran et al., 2024; Zhang et al., 2024).

**Graph Neural Networks.** K-hop Graph Neural Networks (GNNs) (Brossard et al., 2020; Chien et al., 2020; Wang et al., 2020; Abu-El-Haija et al., 2019) are a generalization of the message-passing

type of GNNs. Although most message-passing GNNs iteratively aggregate information from the neighbors to update node representations, K-hop GNNs perform message passing from not only the 1st hop but all the neighbors within K hops of the node, and different weights and activations are used for each hop. K-hop GNNs strictly generalize message-passing GNNs, since an L-layer messaging passing GNN is an L-layer K-hop GNN with K=1. Note that when we say K-hop GNN, we mean that each node v receives messages from all the neighbors that have a distance from node v less than or equal to K. In contrast, k-hop GNNs pass messages from only the neighbors that are exactly distance k from node v. K-hop GNNs have been mathematically shown to be more powerful than K=1 GNNs at encoding the topology of the graph and can distinguish almost all regular graphs (Feng et al., 2023).

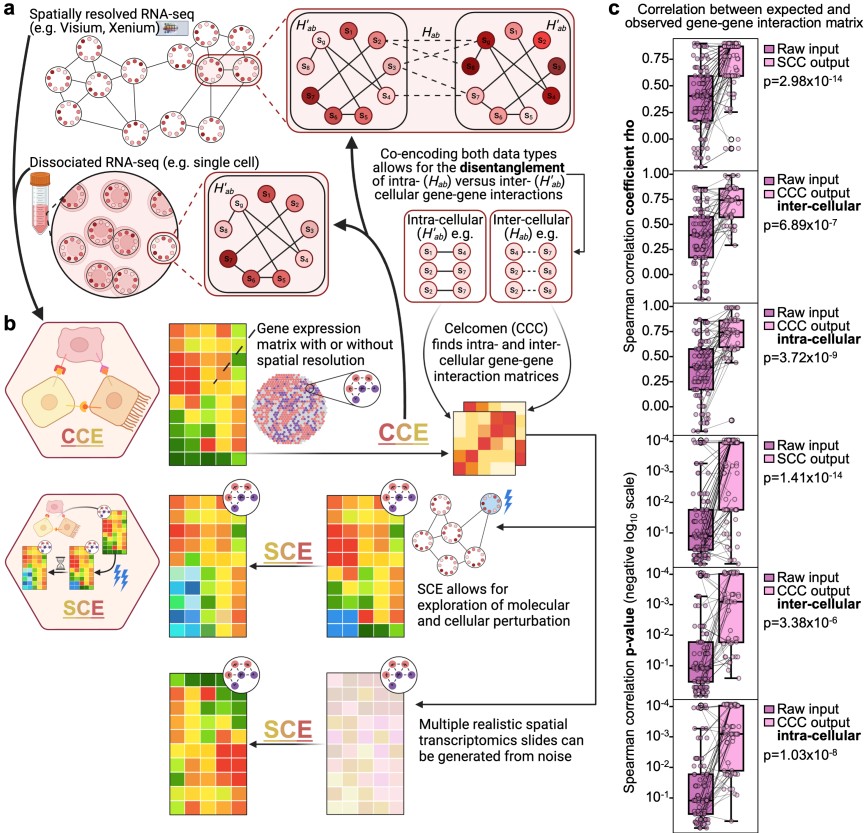

Figure 2: Celcomen reproduces its identifiability guarantees in simulations. a) Inference module (CCE) can learn gene-gene relationships from either spatially resolved and (optionally) dissociated RNA-seq data. The highlighted cell-cell pair, in spatial data, and individual cell, in scRNA-seq data, emphasizes how CCE can distinguish gene-gene interactions that are intra- ($H'_{ab}$) vs. inter- ($H_{ab}$) cellular gene-gene interactions. b) Generative module, called Simcomen (SCE), leverages learned gene-gene relationships from CCE to counterfactual tissue behavior after cellular or genetic perturbation. c) Box plots with the x-axis as the dataset being correlated against the raw noise inputted into SCE to generate data as magenta and the learned output from SCE or the learned gene-gene relationships from CCE as pink. The y-axis depicts the Spearman correlation coefficient rho (upper three) or correlation p-value (lower three). Mann-Whitney U-test p-values are labeled on the center right of each plot and the legend on the upper right of each plot labels each box's dataset.

## 3   METHOD

We now describe the method proposed in this paper, called Celcomen. We start by discussing its motivation and inspiration stemming from Lagrangian mechanics in section 3.1, then the biological assumptions behind Celcomen in Section 3.2, and finally the model's derivation in Section 3.3.

## 3.1 MOTIVATION AND INSPIRATION

Causal inference in machine learning aims to extract causal structures from data. As such, it stands in between correlation-based methods, and mechanistic models (Peters et al., 2017). To distinguish between correlation and causality, let us assume, for instance, that gene 1 in a cell causes gene 2 in its 1-neighboring cells and gene 2 in a cell causes gene 3 in its 1-neighboring cells (see Fig.1C). Since in some cases (see Fig.1D), the 1-neighbor of the 1-neighbor of a cell is also a 1-neighbor (and not a 2-neighbor) of that same cell, there will be (spurious) colocalization also of genes 1 and 3 in nearest neighbors. A causal model should be able to de-confound such spurious connections within spatial correlations, even without mechanistic data such as epigenetic information. The inspiration for our work comes from the notion of force in physics. In broad strokes, we aim to learn the "least" number of forces (i.e., least number of causal mechanisms that can explain all the observed colocalizations of pairs of genes) that can explain the observed spatial correlations of pairs of genes. "Least" here is meant in the sense of entropy, not the absolute number of forces, although, we could additionally impose a $\ell_1$ norm penalty on the force matrix.

In the Lagrangian formulation of classical physics, we think of time evolution of physical objects as an optimization problem (optimizing the action) such that certain constraints imposed by Lagrange multipliers are obeyed. One can show that Lagrange multipliers are equal to the force required to impose the corresponding constraint, which means that they are meaningful, physical quantities (Goldstein, 1980). At the same time, imposing the constraint via the Lagrange multipliers allows us to remain agnostic about the nature of the force (whether it is electromagnetism, gravity, or nuclear forces) that imposes the constraint. For example, for an ant forced to walk on a table's surface, the constraint's force is electromagnetism, but we do not need to know this in advance to calculate its value. Similarly, in single-cell genomics, measurements are valued in a high-dimensional gene expression space, but they often are hypothesized to lie on a lower dimensional surface (Fefferman et al., 2013) due to biological mechanisms (already discovered or not) that "force" our measurements to lie on it. Uncovering such causal links is the first step to identifying the underlying molecular mechanisms. We use Lagrange multipliers to impose the observed colocalization of genes.

Our key idea is to make a graph neural network whose parameters are guaranteed to assume the values of the Lagrange multipliers of the problem at hand. This idea has two powerful implications: (i) Because Lagrange multipliers are meaningful and physical quantities, finding them is likely to be a well-posed problem, leading to a causally identifiable model. (ii) Since they are meaningful quantities they should be easier to recover, thereby providing a new avenue to mechanistic interpretability.

## 3.2 MODEL ASSUMPTIONS

Our model, Celcomen, is the *unique* model that follows from three assumptions:

- The expected gene-gene correlations across 1st neighbors match exactly the observed ones.
- The expected gene-gene correlations within spots/cells match exactly the observed ones.
- Causal sufficiency: there are no unmeasured common causes of any pair of genes that are under consideration, such as different experimental study sites, different tissues which might have more or less signature of long-range hormonal regulation, etc.

These three assumptions can be summarized in the following equation for the entropy

$$
\begin{aligned}
\mathcal{S}(P(\{s_i^\alpha\}), g_{\alpha\beta}, g'_{\alpha\beta}) = & -\sum_{\{s_i^\alpha\}} P(\{s_i^\alpha\}) \log(P(\{s_i^\alpha\})) \\
& + \sum_{\alpha,\beta} g_{\alpha\beta}(\langle \sum_{i,j\ nn} s_i^\alpha s_j^\beta \rangle_P - \langle \sum_{i,j\ nn} s_i^\alpha s_j^\beta \rangle_{\exp}) \\
& + \sum_{\alpha,\beta} g'_{\alpha\beta}(\langle \sum_i s_i^\alpha s_i^\beta \rangle_P - \langle \sum_i s_i^\alpha s_i^\beta \rangle_{\exp}),
\end{aligned} \tag{1}
$$

where $s_\alpha^i$ is the spatial gene expression and $P(\{s_\alpha^i\})$ is the probability distribution over spatial transcriptomics samples, and $g'_{\alpha\beta}$, $g_{\alpha\beta}$ are Lagrange multipliers that enforce our assumptions 1 and 2. For a summary of our notation, see Appendix A, and for a discussion of what the biological limitations and implications of these assumptions see Appendix J. Our task is to maximize the entropy

functional in equation 1 over all possible functions $P \in L^1(\mathbb{R}^{N \times S})$ and matrices $g_{\alpha\beta}$ and $g'_{\alpha\beta}$:

$$\max_{P,g,g'} \mathcal{S}(P(\{s_i^\alpha\}), g_{\alpha\beta}, g'_{\alpha\beta}). \tag{2}$$

### 3.3 MODEL DERIVATION

We should note that the optimization problem in equation 2 is a particularly hard non-parametric problem because it requires optimizing over the space of normalized functions. On a similar note, the entropy is a *functional* that needs to be maximized, not merely a function. To this end, using functional calculus, we perform the maximization of the entropy functional in equation 1 over all functions $P \in L^1(\mathbb{R}^{N \times S})$, to obtain a simpler optimization problem over $g, g'$ alone. This relaxed optimization problem is more amenable to neural networks and will reveal the required architecture for Celcomen. The aim of this section is to *derive* the suitable structure of Celcomen that directly follows from the assumptions of the previous section.

**Theorem 1** (Extremization over $P$). *The following two optimization problems are equivalent*

- *Maximizing the entropy functional in equation 1 over all possible functions $P \in L^1(\mathbb{R}^{N \times S})$ and matrices $g_{\alpha\beta}$ and $g'_{\alpha\beta}$*

$$\max_{P,g,g'} \mathcal{S}(P(\{s_i^\alpha\}), g_{\alpha\beta}, g'_{\alpha\beta}) \tag{3}$$

  *where $\mathcal{S}$ is given by equation 1.*

- *Minimizing the experimental/empirical log-likelihood over matrices $g_{\alpha\beta}$ and $g'_{\alpha\beta}$,*

$$\min_{g,g'} \langle \log P \rangle_{exp} = \min_{g,g'} \left( -\log Z(g_{\alpha\beta}, g'_{\alpha\beta}) + g_{\alpha\beta} C_{\alpha\beta}^{exp} + g'_{\alpha\beta} C_{\alpha\beta}^{'\,exp} \right) \tag{4}$$

  *where $C_{\alpha\beta} = \sum_{i,j} s_j^\alpha J_{ji} s_i^\beta$, $C'_{\alpha\beta} = \sum_i s_i^\alpha s_i^\beta$, $Z = \sum_{s_i^\alpha} e^{\mathcal{H}(\{s_i^\alpha\})}$ and*

$$\mathcal{H} = \sum_{\alpha\beta} \sum_i s_i^\alpha g'_{\alpha\beta} s_i^\beta + \sum_{\alpha\beta} \sum_{i,j} s_i^\alpha J_{ij} g_{\alpha\beta} s_j^\beta. \tag{5}$$

Theorem 1, whose proof is in Appendix B, implies that maximizing $\mathcal{S}$ is equivalent to minimizing

$$\langle \log P \rangle_{\exp} = -\log Z(g_{\alpha\beta}) + g_{\alpha\beta} C_{\alpha\beta}^{\exp} + g'_{\alpha\beta} C_{\alpha\beta}^{'\,\exp}, \tag{6}$$

where $C_{\alpha\beta} = \sum_{i,j} s_j^\alpha J_{ji} s_i^\beta = \mathbf{s}^\top \mathbf{J} \mathbf{s}$, and $g_{\alpha\beta} C_{\alpha\beta}^{\exp} = Tr(\mathbf{s}\mathbf{g}\mathbf{s}^\top \mathbf{J}) = Tr(\mathbf{J}\mathbf{s}\mathbf{g}\mathbf{s}^\top)$, see Appendix A for notation reminders. We note that, the term $\mathbf{J}\mathbf{s}\mathbf{g}$ is a standard linear message passing as in GCN (Kipf & Welling, 2017) and other GNN methods, as shown in (Eliasof et al., 2023)

In summary, our non-parametric optimization over $P$, tells us that the desired model architecture is a k-hop graph convolutional network, similar to (Nikolentzos et al., 2020; Feng et al., 2023) with a new and simpler loss function, Equation 6.

A key challenge in causal structure learning and causal inference is determining whether multiple causal structures could explain the observed data or, more precisely, whether the causal structure is identifiable (i.e., a unique explanation exists) from the data. For a causal model to be well-defined, it must have a unique causal structure that fits the data; otherwise, perturbation effects cannot be identified, and the model may arbitrarily select one of many equally plausible structures due to minor variations in data noise. We now present a theoretical result (proved in Appendix F) that guarantees Celcomen's identifiability in causal structure learning and disentanglement.

**Theorem 2** (Identifiability). *The model defined by $P(\{s_i^\alpha\}|\{g_{\alpha\beta}, g'_{\alpha\beta}\}) = \dfrac{e^{\mathcal{H}(\{s_i^\alpha\})}}{Z}$ is identifiable in the sense that*

$$\forall \{s_i^\alpha\}: \ P(\{s_i^\alpha\}|\{g_{\alpha\beta}, g'_{\alpha\beta}\}) = P(\{s_i^\alpha\}|\{h_{\alpha\beta}, h'_{\alpha\beta}\}), \tag{7}$$

$$\Rightarrow \ g_{\alpha\beta} = h_{\alpha\beta} \text{ and } g'_{\alpha\beta} = h'_{\alpha\beta}. \tag{8}$$

In the following Section, we verify Celcomen's identifiability of causal structure learning and disentanglement in self-consistency synthetic and real-world data experiments.

Celcomen overcomes some limitations of existing methods (see section 2) by leveraging a causally identifiable framework into a generative graph neural network for learning disentangled representations of intra- and inter- cellular gene regulation in spatial transcriptomics data (Fig. 2a-b).
The inference module of Celcomen, hereby called CCE, finds disentangled representations of gene interactions at the cell, 1st neighbor, 2nd neighbor, etc. levels. These representations can then be used by the generative module of Celcomen (Simcomen), hereby called SCE (see Appendix H and Fig. 2b), to produce single-cell spatially resolved predictions of tissue behavior post perturbation and to derive realistic slides of spatial transcriptomics data from noise.

Finally, thanks to carefully deriving powerful approximations to computationally heavy tasks in the training process (see Appendices C, D), Celcomen easily scales to many graphs, cells, and genes, while at the same time having a small number of tunable hyperparameters.

## 4 EXPERIMENTS

We now validate the robustness and insightfulness of CCE and SCE across a plethora of simulations, input types, and human tissues. In summary, we demonstrate Celcomen as a mathematically grounded spatial and single-cell transcriptomics analysis tool that introduces the capability to perform high-resolution spatially resolved perturbation predictions that are critical for clinically relevant disease modeling and tissue engineering efforts.

### 4.1 MATHEMATICAL IDENTIFIABILITY AND INTERPRETABILITY

We now confirm whether Celcomen's identifiability guarantees, discussed in section 3.3, hold in practice by subjecting Celcomen to a multitude of self-consistency experiments.

**Synthetic Problem Setting.** We randomly generate a ground truth set of feature-feature (e.g. gene-gene) interactions, which we encode in the networks weights. Next, we utilize Celcomen's generative module, SCE, to generate spatial transcriptomics data representative of these gene-gene interactions. Following that, we feed the generated data into Celcomen's inference module, CCE, in an attempt to retrieve the originally encoded gene-gene interaction forces, which are also the network weights (see Appendix G for more details).

**Results on synthetic data.** In agreement with its identifiability guarantee, Celcomen consistently demonstrated strong alignment between its inferred gene-gene interactions from its simulated data and the ground truth (Fig. 2c). This result suggests that Celcomen possesses strong self-consistency, and thus identifiability. This conclusion emerges because Celcomen can move between encoded gene-gene interactions to simulated spatial transcriptomics and then back to inferred gene-gene interactions with minimal, if not no, loss of information. It also demonstrates Celcomen's usefulness as a toy model for mechanistic interpretability since its network weights can be reverse-engineered from its observed input-output behavior.

**Results on real-world data.** Furthermore, we confirm Celcomen's identifiability guarantees on real human data, by applying the Celcomen model to multiple spatial transcriptomics slides of human fetal spleen (Suo et al., 2022). For each slide, we trained a sample-specific model and a model trained on the remaining samples. We then correlated the gene-gene interaction matrices of these two models. In line with its claimed identifiability, we observed Spearman correlations in the range of 0.5-0.6 between these two gene-gene interaction matrices even though they shared no training samples, as shown in Fig. 6. Moreover, the gene interactions captured in the intra and inter-cellular matrices are biologically sensible because they adhere to known biological intra and inter-cellular processes, as shown in Fig. 8. Therefore, through this experiment, we demonstrate Celcomen's identifiability, by confirming that its implied stability and robustness extend beyond theory and synthetic data, and can also be observed on real human samples.

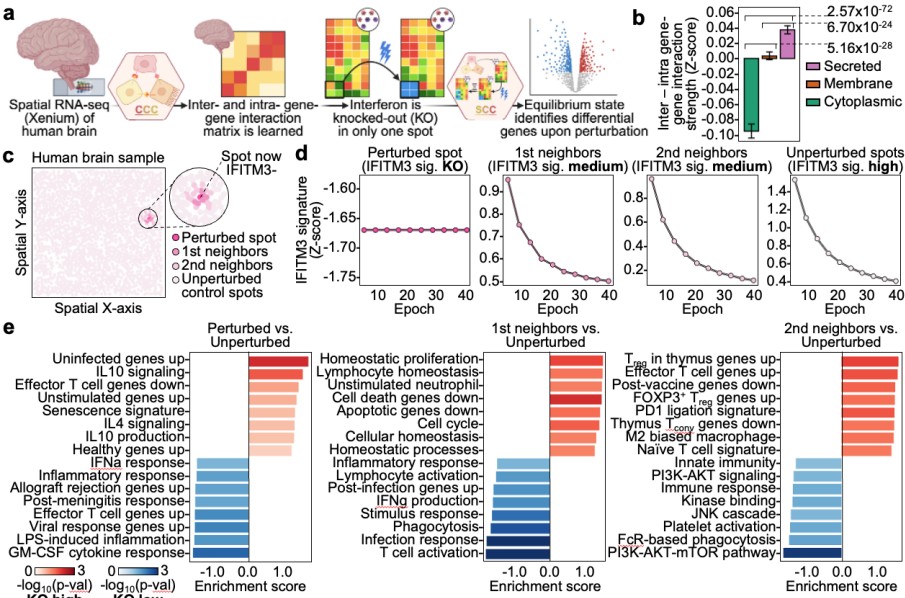

Figure 3: Celcomen recapitulates known interferon knockout biology in human glioblastoma and disentangles intra- and inter-cellular gene-gene interactions. a) Public spatially resolved RNA-seq data, Xenium, of the human brain during glioblastoma is inputted into CCE to derive intra- and inter-cellular gene-gene relationships. Interferon (IFN) signaling is knocked out (KO) in a previously IFN+ cell and SCE learns the local and global effects of this perturbation. b) Bar plot with the x-axis as the subcellular localization of the gene as acquired from its gene ontology and the y-axis is the difference between the gene's inter- and intra-cellular gene-gene interaction terms. c) Visualization of perturbed spot and its nearest neighbors. d) Scatter plots with the x-axis as the epoch number and the y-axis as the interferon signature score of the given spot(s) at the specified epoch. e) Pathway enrichment analysis in each of the groups of cells from (c). Pathways were derived by first calculating pre- and post-perturbation changes in gene expression in each cell, then identifying differentially changed genes between spot(s) of interest and unperturbed controls, this provides a ranking of genes that were differentially upregulated or downregulated in the interferon KO cell, or its neighbors, as compared to the unperturbed control cells.

## 4.2 CAUSAL DISENTANGLEMENT AND SPATIAL COUNTERFACTUALS IN REAL DATA

We now seek to evaluate two other important properties of Celcomen – its ability to disentangle intra-cellular from inter-cellular gene regulation programs, as well as perform spatially resolved perturbation modeling. Disentangling such programs is a crucial requirement of any AI model of a virtual tissue at single-cell resolution, and having such virtual models would enable us to design and test perturbations in silico, thereby accelerating the discovery process in biomedicine (as discussed in section 1). In this section, we focus on counterfactuals involving the knockout of interferon signaling, a crucial pathway in the response of the immune system.

**Disentangling intra- vs inter-cellular programs.** To test the disentanglement abilities, we apply Celcomen in a real, human clinical setting, by analyzing a single-cell resolution spatial transcriptomics dataset of human glioblastoma (brain cancer), as illustrated in Fig. 3a. In congruence with our theoretical understanding of Celcomen, discussed in Thm. 2, we find that Celcomen can successfully disentangle intrinsic from extrinsic sources of transcriptomic variation. This is enabled through Celcomen's assignment of gene-gene interactions involving secreted genes as inter-cellular, and those solely involving cytoplasmic genes as intra-cellular, as shown in Fig. 3b. It is important to note that the knowledge of which genes are secreted and which are cytoplasmic is not encoded into the model as prior information, but rather is learned by the model in an unsupervised manner.

**In silico knockout of interferon in a cell prevents its propagation to neighbors.** We leverage Celcomen's perturbation abilities to model interferon signaling, in the context of a neurological tumor,

where we investigated the scenario of interferon knockout. We choose to model interferon signaling due to its critical role in cancer in inducing antigen presentation, inflammation, and immune activation (Kruse et al., 2023; Haanen, 2013; Gocher et al., 2022). First, we quantified the expression of our sample's interferon-associated gene program by averaging differentially upregulated genes in interferon (IFITM3) high versus low cells. Next, we knocked out interferon expression in a randomly chosen interferon-high cell (Fig. 3c). Utilizing this interferon score, we not only confirmed our in-silico knockout of interferon in the perturbed cell, by observing its marked loss of interferon-associated genes, but we also observed loss of interferon signaling in neighbors of the perturbed cell (Fig. 3d). This behavior is highly consistent with known interferon biology as interferon signaling physically propagates from cell to cell within human tissues; thus, recapitulating this intercellular signaling phenomenon supports the validity of Celcomen's perturbation modeling (Green et al., 2017; Mesev et al., 2019; Lukhele et al., 2019). Moreover, the genes that are active in the intra- and inter-cellular matrices are also associated with processes known to be intra- and inter-cellular, respectively, as illusrated in Fig. 7.

**In silico knock out of interferon in a cell shuts down the immune system in neighbors.** To further confirm the validity of our interferon knockout modeling, we performed pathway enrichment on genes that were differentially changed in perturbed (and perturbed neighboring) compared to unperturbed cells (see Appendix G). Indeed, we find that post-interferon knockout, perturbed cells and their neighbors significantly downregulated characteristic interferon response programs compared to unperturbed cells (Fig. 3e). For example, we observed the perturbed cells to have decreased T cell effector and activation gene programs, as well as greater loss of infection-related gene sets and marked increases in regulatory programs. The consistency of our model with multiple aspects of known interferon biology strongly affirms Celcomen's ability to model perturbations with spatial resolution. Thus, through an in-depth study of Celcomen's application on a real human sample, we provide validation to its value in disentangling intra- versus inter-cellular gene regulation programs and in performing high-resolution spatially contextualized perturbation modeling with accuracy.

## 4.3 VALIDATION OF SPATIAL COUNTERFACTUALS IN-VIVO

To further demonstrate the usefulness of our Celcomen, we now benchmark it on the *in-vivo* full-transcriptome dataset that has measured gene knockouts in spatial transcriptomics, called Perturb-map (Dhainaut et al., 2022), and compare it to a random baseline. To the best of our knowledge, this is the first learning method that attempts to utilize this dataset.
The Perturb-map dataset consists of a mouse model for KP lung cancer where in addition there might be either a Jak2 or a Tgfbr2 knock-out. Their dataset has annotated 5 spatial regions as lesions whose parts are either 1) KP wild-type cancer, or 2) KP cancer with Jak2 knock-out, or 3) KP-cancer with Tgfbr2 knock-out, see Fig. 4a. Notably Perturb-map is based on the Visium technology which is not single-cell resolution because each spot contains cell communities of around seven cells on average.

**Celcomen correctly captures the perturbation effects in the spatial context.** To learn the cancer biology of the KP model, we isolated the wild type parts of the lesions by removing all spots with guide RNAs as well as their 1-neighbor spots. We then applied CCE on this part of the data to learn the gene-gene force matrices of KP cancer biology.

We then chose one of these spots, knocked-out Tgfbr2 (respectively Jak2) in it, and applied SCE to simulate in-silico the effect of this knock-out, see Fig. 4b. Spearman correlation was used to compare model Tgfbr2 KO versus WT gene rankings with those directly derived from experimental Tgfbr2 KO spots and WT which was used as ground truth. We derived "random" controls for each lesion by computing correlations on shuffled gene rankings of the observed and predicted differentials between Tgfbr2 KO and WT. Mann-Whitney U test is used to derive p-value when comparing observed lesion-derived gene rankings with those from random shuffling. Our results show Spearman correlations between our predictions and in-vivo measurements in the range of 0.28-0.47 for all lesions. To assess the significance of this performance, we compare our model to a random baseline where Celcomen is run on randomly shuffled data. We find that Celcomen's performance is significantly higher, with a p-value of 0.0079, compared to the random baseline, as shown in Fig. 4c-f.

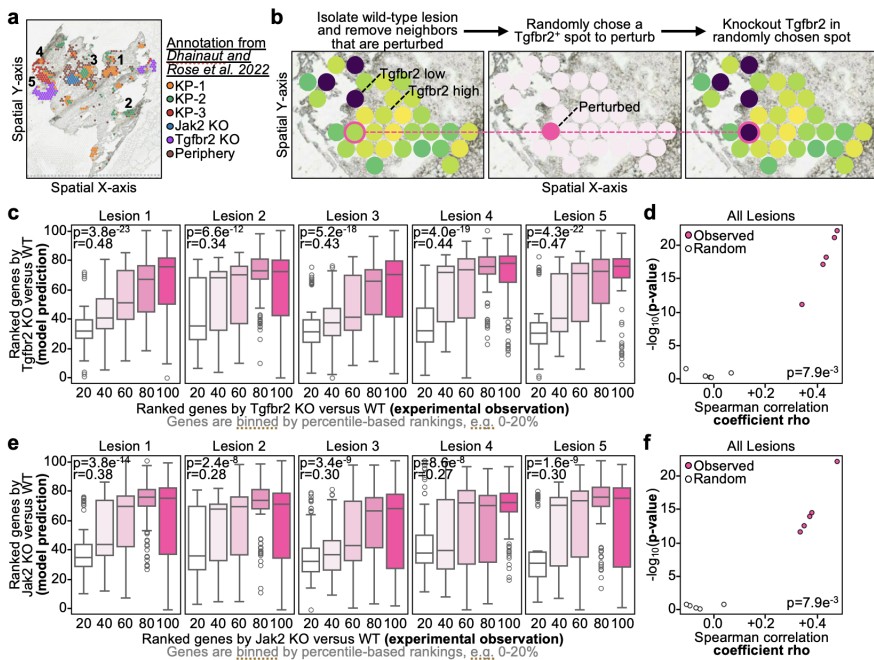

Figure 4: Counterfactual predictions validated in-vivo in lung cancer model. a) Scatter plot on spatial axes with each Visium spot colored by tumor cell phenotype ("KO" for knockout and "KP" for wild-type tumors). Lesions of interest, large enough for modeling, are labeled with numbers. b) Example workflow with a wild type (WT) lesion trimmed for spots within two- degrees of perturbed clusters, random Tgfbr2+ spot has Tgfbr2 knocked out, Celcomen then predicts the whole transcriptome accompanying this perturbation. c,e) Box plots, per lesion, with x-axis as the observed ranked differentially expressed genes (DEGs) between Tgfbr2 (respectively Jak2) KO and WT, and the y-axis as the model predicted gene ranking between our perturbed Tgfbr2 KO spot and wild type spots. Spearman correlation coefficient rhos and p-values are annotated on the plot. d,f) Scatter plot with each dot representing a given tumor lesion with Tgfbr2 (respectively Jak2) KO and the x-axis as the Spearman correlation coefficient rho and y-axis as the p-value, the color indicates if the correlation was computed on the lesion's observed gene rankings or a random shuffling of the gene rankings. Mann-Whitney U test p-value between observed and randomly shuffled correlations are annotated.

## 5 CONCLUSIONS AND DISCUSSION

The advent of single-cell resolution spatial transcriptomics has allowed spatial tissue atlases with unprecedented resolution (Farah et al., 2024; Megas et al., 2024a; To et al., 2024; Yao et al., 2023; Lindeboom et al., 2024; Zhang et al., 2023a). Current computational methods addressing spatial transcriptomics focus on phenotypic characterization but often neglect combined cell- and tissue-level causal perturbation modeling (Rood et al., 2024) which could reveal the mechanisms behind tissue disease states. Here, we present a first step towards a model of Virtual Tissues, called Celcomen, which can predict the effect of spatial counterfactuals at the cellular and tissue levels thanks to its strong mathematical foundations. We confirm Celcomen's ability to disentangle and recover ground truth gene-gene interactions in real and self-simulated spatial transcriptomics data. Moreover, our model opens a new route to mechanistic interpretability via causal inference. As we demonstrate in experiments, thanks to our model's causal identifiability, we can recover the values of the parameters of the neural network with high accuracy.

**Broader Impact.** Celcomen's advancements are poised to significantly impact biomedicine by revealing how diseases cause tissue failure and enabling testable hypotheses on therapeutic benefits. As technology progresses, Celcomen's value will grow, enhancing disease modeling and mechanistic understanding. Additionally, its architecture holds promise for advancing mechanistic interpretability, contributing to the development of causal foundation models.

## REPRODUCIBILITY STATEMENT

Our source code for the method is available at https://shorturl.at/cNQt0 and the code for reproducing the results of our experiments on synthetic and real data is available at https://shorturl.at/js43t. The real data can be downloaded from the links provided in Appendix G.1.

We also provide clear explanations and step-by-step proofs of all the mathematical claims in the paper. Thm. 1 on rewriting our main optimization problem is proved in Appendix B, and Thm. 2 on the idenfiability of our model is proved in Appendix F. Several more technical lemmas on deriving the mean-field approximation used in Celcomen are proved in Appendices C, D.

## ETHICAL STATEMENT

In this work, we do not release any datasets or models that could be misused, and we believe our research carries no direct or indirect negative societal implications. We do not work with sensitive or privacy-related data, nor do we develop methods that could be applied to harmful purposes. To the best of our knowledge, this study raises no ethical concerns or risks of negative impact. Additionally, all human samples used in this work were downloaded from appropriately licensed sources (see Reproducibility Statement 5). We also confirm that there are no conflicts of interest or external sponsorships influencing the objectivity or results of this study.

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

## A  APPENDIX: NOTATION

For ease of reference, we provide here a guide to our notation choices:

- $s_i^\alpha \in \mathbb{R}^{N \times S}$, count values for spot/cell $i$ and gene $\alpha$, where $\forall i : \|s_i\| = 1$,
- $\mathcal{H}$, Hamiltonian of a system,
- $Z = \sum_{\{s_i^\alpha\}} e^{\mathcal{H}(\{s_i^\alpha\})}$, the partition function,
- $\sum_{<i,j>nn}$, sum over pairs of nodes $\{i, j\}$ that are nearest neighbors,
- q, the number of nearest neighbors (that we assume are interacting),
- $\mathcal{S}$, the entropy functional,
- S, the number of spots/nodes in the spatial graph,
- N, the number of features/genes in the graph,
- $g_{\alpha\beta} \in \mathbb{R}^{N \times N}$, Lagrange multiplier enforcing gene-gene correlations,
- $J_{ij} \in \mathbb{R}^{S \times S}$, the spatial adjacency matrix between spots/nodes in the graph,
- $\langle\rangle_P$, the average with respect to the probability distribution $P$,
- $\langle\rangle_{\text{exp}}$, the empirical/experimental average with respect to the observed samples,
- $P(s_i^\alpha) \in L^1(\mathbb{R}^{N \times S})$, the probability density of the count matrix of a spatial, transcriptomics experiment equals the matrix $s_i^\alpha$.

## B  APPENDIX: EXTREMIZATION OVER $P$

Here we prove Theorem 1 which states that the following two optimization problems are equivalent:

- Maximizing the entropy functional in eq 1 over all possible functions $P \in L^1(\mathbb{R}^{N \times S})$ and matrices $g_{\alpha\beta}$ and $g'_{\alpha\beta}$

$$\max_{P,g,g'} \mathcal{S}(P(\{s_i^\alpha\}), g_{\alpha\beta}, g'_{\alpha\beta}) \tag{9}$$

  where $\mathcal{S}$ is given by 1,
- Minimizing the experimental/empirical log likelihood over matrices $g_{\alpha\beta}$ and $g'_{\alpha\beta}$

$$\min_{g,g'} \langle \log P \rangle_{\text{exp}} = \min_{g,g'} \left( -\log Z(g_{\alpha\beta}, g'_{\alpha\beta}) + g_{\alpha\beta} C_{\alpha\beta}^{\text{exp}} + g'_{\alpha\beta} C_{\alpha\beta}^{'\text{exp}} \right) \tag{10}$$

  where

$$C_{\alpha\beta} = \sum_{i,j} s_j^\alpha J_{ji} s_i^\beta, \tag{11}$$

$$C'_{\alpha\beta} = \sum_i s_i^\alpha s_i^\beta, \tag{12}$$

$$Z = \sum_{s_i^\alpha} e^{\mathcal{H}(\{s_i^\alpha\})}, \tag{13}$$

$$\mathcal{H} = \sum_{\alpha\beta} \sum_i s_i^\alpha g'_{\alpha\beta} s_i^\beta + \sum_{\alpha\beta} \sum_{i,j} s_i^\alpha J_{ij} g_{\alpha\beta} s_j^\beta. \tag{14}$$

*Proof.* Optimizing a functional requires taking derivatives with respect to functions. In particular, using $\frac{\delta \int f(x)dx}{\delta f(y)} = \delta(x - y)$, we can maximize $S$ with respect to $P$:

$$0 = \frac{\delta \mathcal{S}}{\delta P(s')} = -\log P(\{s'\}) - 1 + \sum_{\alpha,\beta} g_{\alpha\beta} \sum_{i,j\, nn} s_i^{'\alpha} s_j^{'\beta} + \sum_{\alpha,\beta} g'_{\alpha\beta} \sum_i s_i^{'\alpha} s_i^{'\beta} \tag{15}$$

$$\Rightarrow P(\{s_i^\alpha\}|\{g'_{\alpha\beta}, g_{\alpha\beta}\}) = \frac{e^{\mathcal{H}(\{s_i^\alpha\})}}{Z} \tag{16}$$

where we normalized the probability function and denote

$$\mathcal{H} = \sum_{\alpha\beta}\sum_i s_i^\alpha g'_{\alpha\beta} s_i^\beta + \sum_{\alpha\beta}\sum_{<i,j>nn} s_i^\alpha g_{\alpha\beta} s_j^\beta \tag{17}$$

$$= \sum_{\alpha\beta}\sum_i s_i^\alpha g'_{\alpha\beta} s_i^\beta + \sum_{\alpha\beta}\sum_{i,j} s_i^\alpha J_{ij} g_{\alpha\beta} s_j^\beta, \tag{18}$$

$$Z = \sum_{s_i^\alpha} e^{\mathcal{H}(\{s_i^\alpha\})}. \tag{19}$$

Maximizing with respect to the Lagrange multipliers $g_{\alpha\beta}$, $g'_{\alpha\beta}$ gives:

$$0 = \langle \sum_{i,j\ nn} s_i^\alpha s_j^\beta \rangle_P - \langle \sum_{i,j\ nn} s_i^\alpha s_j^\beta \rangle_{\exp}, \tag{20}$$

$$0 = \langle \sum_i s_i^\alpha s_i^\beta \rangle_P - \langle \sum_i s_i^\alpha s_i^\beta \rangle_{\exp}. \tag{21}$$

Moreover, by substituting 16 into 1 we get

$$\mathcal{S}(P(\{s_i^\alpha\}), g_{\alpha\beta}, g'_{\alpha\beta}) = \log Z - g_{\alpha\beta}\langle \sum_{i,j\ nn} s_i^\alpha s_j^\beta \rangle_{\exp} - g'_{\alpha\beta}\langle \sum_i s_i^\alpha s_i^\beta \rangle_{\exp} \tag{22}$$

$$= -\langle \log P(s) \rangle_{\exp} \tag{23}$$

Therefore maximizing $\mathcal{S}$ is equivalent to minimizing

$$\langle \log P \rangle_{\exp} = -\log Z(g_{\alpha\beta}) + g_{\alpha\beta} C_{\alpha\beta}^{\exp} + g'_{\alpha\beta} C'^{\exp}_{\alpha\beta} \tag{24}$$

where $C_{\alpha\beta} = \sum_{i,j} s_j^\alpha J_{ji} s_i^\beta$.

$\square$

## C APPENDIX: MEAN GENE APPROXIMATION

Even after performing the optimization over functions $P$, the simpler loss function 4 of our k-hop GCN is still intractable to compute because calculating the partition function (and its derivatives) requires summing over a large number of possible spatial transcriptomics datasets.

Several famous algorithms in machine learning circumvent computing the partition function in different ways. For instance, contrastive learning essentially takes the ratio of probabilities, thereby canceling out the partition function; score-based diffusion uses score-matching to learn a model of the gradient of the log of the probability density function (Song et al., 2021), which again avoids computing the partition function.

In this paper, we introduce a novel approximation to the partition function, inspired by physics Gardiner & Megas (2021); Kraus et al. (2020), which has never been used before in spatial transcriptomics. This is a new Mean Field Theory approximation

$$s_k^\alpha = \bar{s}_k^\alpha + \delta s_k^\alpha = m^\alpha + \delta s_k^\alpha \tag{25}$$

where we assume that the gene expression does not fluctuate too much around the mean.

**Lemma 1** (Mean Field Theory approximation). *Using the mean field theory approximation, $s_k^\alpha = \bar{s}_k^\alpha + \delta s_k^\alpha = m^\alpha + \delta s_k^\alpha$ and retaining terms of order $\mathcal{O}(\delta s_k^\alpha)$, the partition function assumes the form*

$$Z = \sum_{s_i^\alpha} e^{\mathcal{H}(\{s_i^\alpha\})}, \tag{26}$$

$$\mathcal{H} = \sum_i \sum_{\alpha,\beta} \left(g'_{\alpha\beta} + \frac{q}{2} g_{\alpha\beta}\right)(-m^\alpha m^\beta + m^\beta s_i^\alpha + m^\alpha s_i^\beta) \tag{27}$$

*Proof.* Our Mean Field Theory approximation is

$$s_k^\alpha = \bar{s}_k^\alpha + \delta s_k^\alpha = m^\alpha + \delta s_k^\alpha \tag{28}$$

where we assume that the gene expression does not fluctuate too much around the mean.

Using this, we can rewrite the exponent as

$$s_i^\alpha g_{\alpha\beta} s_j^\beta = g_{\alpha\beta}(\bar{s}_i^\alpha + \delta s_i^\alpha)(\bar{s}_j^\beta + \delta s_j^\beta) \tag{29}$$

$$\approx g_{\alpha\beta}(\bar{s}_i^\alpha \bar{s}_j^\beta + \bar{s}_j^\beta \delta s_i^\alpha + \bar{s}_i^\alpha \delta s_j^\beta) \tag{30}$$

$$= g_{\alpha\beta}(m^\alpha m^\beta + m^\beta(s_i^\alpha - m^\alpha) + m^\alpha(s_j^\beta - m^\beta)) \tag{31}$$

$$= g_{\alpha\beta}(-m^\alpha m^\beta + m^\beta s_i^\alpha + m^\alpha s_j^\beta) . \tag{32}$$

where in the second line we used the MFT approximation to neglect terms of order higher than 2, and

$$s_i^\alpha g_{\alpha\beta}' s_i^\beta = g_{\alpha\beta}'(-m^\alpha m^\beta + m^\beta s_i^\alpha + m^\alpha s_i^\beta) . \tag{33}$$

This implies that the intercellular term in the exponent can be rewritten as

$$\sum_{\langle i,j \rangle} \sum_{\alpha,\beta} g_{\alpha\beta}(-m^\alpha m^\beta + m^\beta s_i^\alpha + m^\alpha s_j^\beta) = \frac{q}{2} \sum_i \sum_{\alpha,\beta} g_{\alpha\beta}(-m^\alpha m^\beta + m^\beta s_i^\alpha + m^\alpha s_i^\beta) \tag{34}$$

where $q$ is the number of nearest neighbors that we assume are interacting, and therefore

$$\mathcal{H} = \frac{q}{2} \sum_i \sum_{\alpha,\beta} g_{\alpha\beta}(-m^\alpha m^\beta + m^\beta s_i^\alpha + m^\alpha s_i^\beta) \tag{35}$$

$$+ \sum_i \sum_{\alpha,\beta} g_{\alpha\beta}'(-m^\alpha m^\beta + m^\beta s_i^\alpha + m^\alpha s_i^\beta) \tag{36}$$

$$= \sum_i \sum_{\alpha,\beta} \left(g_{\alpha\beta}' + \frac{q}{2} g_{\alpha\beta}\right)(-m^\alpha m^\beta + m^\beta s_i^\alpha + m^\alpha s_i^\beta) \tag{37}$$

since $g_{\alpha,\beta}$ is symmetric.

$\square$

## D    APPENDIX: COMPUTING THE PARTITION FUNCTION IN THE MFT APPROXIMATION

To sum the exponential of the Hamiltonian over all possible count matrices $s_i^\alpha$ we first prove the following Lemma.

**Lemma 2.** *The following sum can be simplified as follows*

$$\sum_{\{s_i^\alpha\}} exp\left[\sum_i \sum_{\alpha,\beta} (\frac{q}{2} g_{\alpha\beta})(m^\beta s_i^\alpha + m^\alpha s_i^\beta)\right] = V_{\mathbb{S}^{n-1}} \left(\frac{e^{qH/2} - e^{-qH/2}}{qH/2}\right)^S \tag{38}$$

*where $S$ is the number of spots, $H_{\alpha\beta} = g_{\alpha\beta} + g_{\beta\alpha}$, $H = \sqrt{\sum_\beta (\sum_\alpha H_{\alpha\beta} m^\alpha)^2}$, and $V_{\mathbb{S}^{n-1}}$ is the volume of the n dimensional sphere.*

*Proof.*

$$Z = \sum_{\{s_i^\alpha\}} \exp\left[\frac{q}{2}\sum_i \sum_{\alpha,\beta} g_{\alpha\beta}(m^\beta s_i^\alpha + m^\alpha s_i^\beta)\right] \tag{39}$$

$$= \sum_{\{s_i^\alpha\}} \exp\left[\frac{q}{2}\sum_i \sum_{\alpha,\beta} (g_{\beta\alpha}m^\alpha s_i^\beta + g_{\alpha\beta}m^\alpha s_i^\beta)\right] \tag{40}$$

$$= \sum_{\{s_i^\alpha\}} \exp\left[\frac{q}{2}\sum_i \sum_{\alpha,\beta} (g_{\beta\alpha} + g_{\alpha\beta})m^\alpha s_i^\beta\right] \tag{41}$$

$$= \sum_{\{s_i^\alpha\}} \exp\left[\frac{q}{2}\sum_i \sum_{\alpha,\beta} H_{\alpha\beta}m^\alpha s_i^\beta\right] \tag{42}$$

$$= \prod_i \left(\int_{s_i \in \mathbb{S}^n} ds_i\right) \exp\left[\frac{q}{2}\sum_i \sum_{\alpha,\beta} H_{\alpha\beta}m^\alpha s_i^\beta\right] \tag{43}$$

$$= \prod_i \left(\int_{s_i \in \mathbb{S}^n} \exp\left[\frac{q}{2}\sum_i H s_i^1\right] ds_i\right) \tag{44}$$

$$= V_{\mathbb{S}^{n-1}} \prod_i \left(\int_0^\pi \exp\left[\frac{q}{2}\sum_i H\cos\theta\right]\sin\theta d\theta\right) \tag{45}$$

$$= V_{\mathbb{S}^{n-1}} \prod_i \left(\int_{-1}^1 \exp\left[\frac{q}{2}\sum_i Hu\right] du\right) \tag{46}$$

$$= V_{\mathbb{S}^{n-1}} \left(\frac{e^{qH/2} - e^{-qH/2}}{qH/2}\right)^S \tag{47}$$

$$\tag{48}$$

where $S$ is the number of spots, $H_{\alpha\beta} = g_{\alpha\beta} + g_{\beta\alpha}$, $H = \sqrt{\sum_\beta (\sum_\alpha H_{\alpha\beta}m^\alpha)^2}$, and without loss of generality we assumed that the vector $\sum_\alpha H_{\alpha\beta}m^\alpha$ lies only along the first dimension. $\square$

Now applying lemma 1 to the results of lemma 2, where we just need to replace $g_{\alpha\beta} \rightarrow g_{\alpha\beta} + \frac{2}{q}g'_{\alpha\beta}$, gives

$$\log Z = -S\sum_{\alpha,\beta}\left(g'_{\alpha\beta} + \frac{q}{2}g_{\alpha\beta}\right)m^\alpha m^\beta$$
$$+ \log V_{\mathbb{S}^{n-1}}$$
$$+ S\log\frac{e^{H'/2} - e^{-H'/2}}{H'/2} \tag{49}$$

where $S$ is the number of spots, $H'_{\alpha\beta} = qg_{\alpha\beta} + qg_{\beta\alpha} + 2g'_{\alpha\beta} + 2g'_{\beta\alpha}$, $H' = \sqrt{\sum_\beta (\sum_\alpha H'_{\alpha\beta}m^\alpha)^2}$

Using 4, 49, we have a complete formula for calculating the log-likelihood, and the only optimization remaining is over the Lagrange multipliers.

$$0 = \frac{\delta P(\{s_i^\alpha\})}{\delta g_{\alpha\beta}} \tag{50}$$

$$0 = \frac{\delta P(\{s_i^\alpha\})}{\delta g'_{\alpha\beta}} \tag{51}$$

In other words, we want to look for the forces that are causing the observed spatial gene expression. Since the Lagrange multipliers/forces are meaningful physical variables, they naturally equip our model with identifiability as we show in the next section.

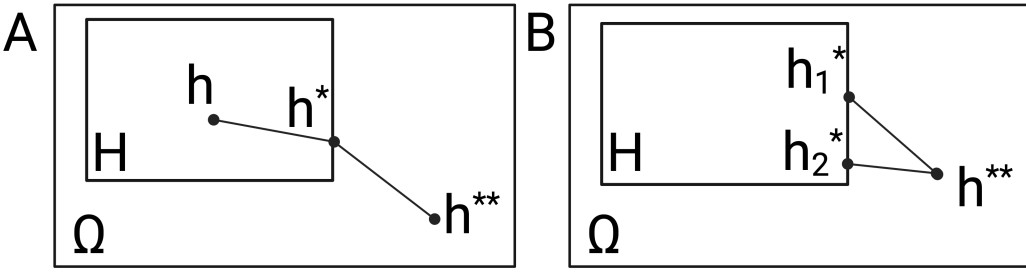

Figure 5: A) An illustration of the decomposition of errors in a PAC model. The model explores the fitness of a subset of hypotheses $\mathbb{H}$. B) A non-identifiable model has many hypotheses that have the smallest approximation error. This can lead to non-robust results as a small change in the data can cause the model to transition to an alternative hypothesis.

## E  APPENDIX: REVIEW OF PROBABLY APPROXIMATELY CORRECT LEARNING

Here we offer a brief review of Probably Approximately Correct (PAC) learning, to set up some notation and definitions that will aid our discussion of identifiability. For a more in-depth discussion of the relevant theory, readers can consult (Mohri et al., 2018).

In PAC learning, we aim to learn the hypothesis $h$ from a set of available hypotheses $\mathbb{H} \subset \Omega$ which has the smallest generalization error $R(h)$ to a given concept $c$. Let $R^* = R(h^{**})$ be the smallest error achievable for any hypothesis in $\Omega$, we call $R^*$ the Bayes error and $h^{**}$ the Bayes optimal hypothesis. Then the difference between the error of any hypothesis $h$ and the Bayes error can be decomposed as

$$R(h) - R(h^{**}) = \underbrace{\left(R(h) - R(h^*)\right)}_{\text{estimation error}} - \underbrace{\left(R(h^*) - R(h^{**})\right)}_{\text{approximation error}}, \tag{52}$$

where $h^* \in \mathbb{H}$ is the hypothesis in $\mathbb{H}$ that minimizes the error, see Fig. 5.

There are examples of famous bounds on the overall error which are often referred to as oracle identities but this requires extra assumptions about the distribution of the data or expected model. For instance, the Lasso oracle inequality is proved assuming the restricted eigenvalue condition (Wainwright, 2019); and the oracle for graphical Lasso is proved assuming that the precision matrix is $\alpha - $ spiky (Wainwright, 2019).

In the context of PAC learning, the identifiability of a model is the property of having a unique hypothesis $h^* \in \mathbb{H}$ that minimizes the approximation error,

$$\exists! h^* \in \mathbb{H} : \forall h \in \mathbb{H} : \ \|R(h^*) - R(h^{**})\| \leq \|R(h) - R(h^{**})\|. \tag{53}$$

Lack of identifiability can lead to model results that are not robust since tiny differences in the data or random seeds could lead the model to transition to one of the other equally optimal hypotheses.

## F  APPENDIX: IDENTIFIABILITY RESULT

An important question we want to address is the identifiability of our model, ie whether there is a unique setting of the forces that leads to the observed correlations in the data. If the identifiability property holds then our model will naturally be robust and causal in the sense that it can deconfound spurious correlations.

In gauge theory terminology, we want to determine whether there is some gauge symmetry that allows different sets of parameters to give the same probability distribution.

We now prove one of the main results of our paper, Theorem 2, which states that the model defined by equation 16 is identifiable since

$$\forall \{s_i^\alpha\}: \ P(\{s_i^\alpha\}|\{g_{\alpha\beta}, g'_{\alpha\beta}\}) = P(\{s_i^\alpha\}|\{h_{\alpha\beta}, h'_{\alpha\beta}\}) \tag{54}$$

$$\Rightarrow \ g_{\alpha\beta} = h_{\alpha\beta} \ \text{and} \ g'_{\alpha\beta} = h'_{\alpha\beta} \tag{55}$$

*Proof.* Let's pick $i$ to be a cell/node that has at least one neighbor. If there is not such a cell then there wouldn't be a cell communication problem to model.

$$P(\{s_i^\alpha\}|\{g_{\alpha\beta}, g'_{\alpha\beta}\}) = P(\{s_i^\alpha\}|\{h_{\alpha\beta}, h'_{\alpha\beta}\}) \tag{56}$$

$$\Rightarrow \ \frac{dP(\{s_i^\alpha\}|\{g_{\alpha\beta}, g'_{\alpha\beta}\})}{ds_i^\alpha} = \frac{dP(\{s_i^\alpha\}|\{h_{\alpha\beta}, h'_{\alpha\beta}\})}{ds_i^\alpha} \tag{57}$$

Then we pick $j$ to be any of the neighbors of cell $i$,

$$\frac{d^2 P(\{s_i^\alpha\}|\{g_{\alpha\beta}, g'_{\alpha\beta}\})}{ds_j^\beta ds_i^\alpha} = \frac{dP(\{s_i^\alpha\}|\{h_{\alpha\beta}, h'_{\alpha\beta}\})}{ds_j^\beta ds_i^\alpha} \tag{58}$$

$$\Rightarrow \ \frac{d^2 P(\{s_i^\alpha\}|\{g_{\alpha\beta}, g'_{\alpha\beta}\})}{ds_j^\beta ds_i^\alpha}\bigg|_{s_{nn\ j}^\beta=0, s_i^\alpha=0} = \frac{dP(\{s_i^\alpha\}|\{h_{\alpha\beta}, h'_{\alpha\beta}\})}{ds_j^\beta ds_i^\alpha}\bigg|_{s_{nn\ j}^\beta=0, s_i^\alpha=0} \tag{59}$$

$$\Rightarrow \ g_{\alpha\beta} = h_{\alpha\beta} \tag{60}$$

Alternatively, taking the second derivative with respect to the same cell $i$,

$$\frac{d^2 P(\{s_i^\alpha\}|\{g_{\alpha\beta}, g'_{\alpha\beta}\})}{ds_i^\beta ds_i^\alpha} = \frac{dP(\{s_i^\alpha\}|\{h_{\alpha\beta}, h'_{\alpha\beta}\})}{ds_i^\beta ds_i^\alpha} \tag{61}$$

$$\Rightarrow \ \frac{d^2 P(\{s_i^\alpha\}|\{g_{\alpha\beta}, g'_{\alpha\beta}\})}{ds_i^\beta ds_i^\alpha}\bigg|_{s_i^\beta=0, s_i^\alpha=0} = \frac{dP(\{s_i^\alpha\}|\{h_{\alpha\beta}, h'_{\alpha\beta}\})}{ds_i^\beta ds_i^\alpha}\bigg|_{s_i^\beta=0, s_i^\alpha=0} \tag{62}$$

$$\Rightarrow \ g'_{\alpha\beta} = h'_{\alpha\beta} \tag{63}$$

$\square$

Identifiability for a model means that a unique configuration of our model fits the observed data. This is very important for causal models since otherwise, our model could tell us that A either causes B or it doesn't cause B, which is an unhelpful tautology. Identifiability is a point of principle of whether the model could definitively decide that A causes B if provided enough data. For this reason most proofs of identifiability happen in the infinite data limit.

## G  Appendix: Methods

### G.1  Spatial transcriptomics dataset curation and preprocessing

The fetal spleen datasets were curated from https://developmental.cellatlas.io/fetal-immune in log-normalized form, which explicitly indicates log-transformation and library size normalization18. The glioblastoma dataset was curated from 10x genomics at https://www.10xgenomics.com/datasets/ffpe-human-brain-cancer-data-with-human-immuno-oncology-profiling-panel-and-custom-add-on-1-standard and subjected to the same library size normalization, counts per million (CPM), and log-transformation, with a base of e; additionally, only genes that were expressed in at least 100 cells were kept. Due to the large size of the Xenium slide, a random square portion of the slide was chosen for analysis, this section is defined as cell centroid x-component $> 6500$ and $< 7000$ and cell centroid y-component $> 8000$ and $< 8500$. The entire fetal spleen slide was kept for each fetal slide sample as they are comparatively smaller than the original Xenium slide and post down-sampling, approximately the same size as the analyzed Xenium section. All data normalization was done using Scanpy (v1.9.8) in Python (v3.9.18)31.

### G.2 SIMULATIONS TESTING CELCOMEN'S IDENTIFIABILITY GUARANTEES

Simulations were done in Python and completed by first generating a random n-genes by n-genes matrix of ground truth gene-gene interactions, for these experiments, four genes were used. We then utilized Celcomen's generative module to create a simulated spatial transcriptomics slide whose values were then fed into Celcomen's inference module to decipher back gene-gene interactions. Spearman correlation was used to compare the original ground-truth gene-gene interaction values and the simulated-then-inferred gene-gene interaction values to test for model robustness and identifiability. For all exact parameter values utilized during the experiments, see the "analysis.simulations.ipynb" notebook in the reproducibility GitHub.

### G.3 BIOLOGICAL TESTING OF CELCOMEN'S IDENTIFIABILITY GUARANTEES

Biological confirmation of Celcomen's identifiability guarantee was done by training two Celcomen inference module instances at the same time and comparing their derived gene-gene interaction results. The first model instance, which we call sample-specific, was trained only on one sample. The second model instance, which we call rest, was training on the remaining samples. Thus, these two model instances are never trained on the same samples. Each model is trained to completion utilizing the same model hyperparameters, and their gene-gene interaction matrices are retrieved after the final epoch. We correlate a flattened version of their gene-gene interaction matrices using Spearman's correlation due to the possible non-linear nature of the matrices' values. We repeat this experiment for each of the samples in the fetal spleen dataset. The results across each sample's experiments are aggregated together and compared in a bar plot. We derived a "random" control to compare to by shuffling the order of the flattened gene-gene interaction matrices and computing a correlation of the shuffled values. Mann-Whitney U test is used to derive p-values and all p-values are labeled on plot. For the full code utilized, see the "analysis.biological.ipynb" notebook in the reproducibility GitHub.

### G.4 INTERFERON KNOCKOUT EXPERIMENT ON XENIUM OF HUMAN GLIOBLASTOMA

Processed Xenium data was subjected to the inference module of Celcomen, CCE, and then these gene-gene interaction values were annotated as containing cytoplasmic, surface membrane (plasma membrane GO ID via GO cellular component), or secreted (extracellular space GO ID also via GO cellular component) genes according to their GO IDs from QuickGO32. IFITM3 was knocked out in a randomly selected previously IFITM3 positive cell. First neighbors were defined as less than 15 μm away and second neighbors were defined as less than 30 μm away. Changes in each gene's expression in each cell were calculated and these changes in expression pre- and post- perturbation were compared between different specified cellular subsets. These are the differential genes later used for differential expression analysis and pathway enrichment. Gene set enrichment analysis (GSEA) in R (v4.1.2) was utilized to perform pathway enrichment analysis on differentially post-perturbation affected genes. The interferon signature was derived directly from tissue by computing the differentially expressed genes between interferon high and low cells and taking the top 25, excluding the perturbed IFITM3 as that would bias analyses. For the full model parameters and code utilized, see the "analysis.perturbation.ipynb" notebook in the reproducibility GitHub.

## H APPENDIX: SIMCOMEN, GENERATION MODULE

Our model is a mathematically robust way of learning the distribution of spatial transcriptomics samples such that there is a 1-1 correspondence between a configuration of forces and the learned distribution of spatial transcriptomics samples.

Generating new samples from the learned distribution is a classic problem that can be addressed, for instance, by Markov Chain Monte Carlo Methods. However, given the high dimensionality of the space of spatial transcriptomics, MCMC can be very computationally expensive. Therefore, in our generation module, called Simulated Communication Energy, we produce new samples in an adversarial approach by trying to find a sample that would trick Celcomen into thinking it is derived from the learned distribution. More specifically, we fix the parameters of our model and optimize the likelihood of the possible datasets. This approach is also followed for generating counterfactual

samples, eg we intervene on a node, and from that starting point we find the most likely sample under the learned distribution of spatial transcriptomics.

## I  COMPUTATIONAL EFFICIENCY

Our derivation of the mean field approximation eq. 49 allows to speed the computation of Z, the partition function. In particular, with eq. 26, which is the un-approximated formula that derives from Theorem 1, computing Z would require summing all possible spatial transcriptomics samples, which we estimate to be roughly equal to $10^{6,000,000}$ possibilities (for 100 possible gene values, 1000 genes, and 3000 spots). Performing this calculation in every training step is computationally infeasible. On the other hand, using eq. 49, which describes our Celcomen, Z can be calculated by summing over $10^6$ gene-pairs. Using this approach, Celcomen's *inference* module runs in 11 seconds for 500 genes and 2500 spots. In the early phases of the design of the model, we experimented with sampling from a space of $10^{6,000,000}$ possibilities using MCMC or Gibbs sampling, which proved to be computationally intractable. That is the reason we implemented the generative module of Celomen to generate the most likely spatial transcriptomics sample based on its learned distribution of observed samples. Thanks to this, running the generative module for 500 genes and 2500 spots takes a few hours on CPU or 12 minutes on a GPU.

## J  LIMITATIONS AND FUTURE DIRECTIONS

While this work demonstrates the potential of Celcomen in disentangling intra- and inter-cellular regulation, identifying causal links, and performing spatial counterfactuals, several limitations highlight exciting directions for future investigation.

Limitation 1: Celcomen aims to identifiably retrieve the undirected form of the DAG underlying intra- and inter-cellular regulation from observational data, i.e., without interventional data. Without interventional data, it is impossible to uncover more than the Markov equivalence class of the DAG, which is the undirected form of the graph along with partial information about colliders. Therefore Celcomen's ability is close to being tight against the best possible bound on the abilities of unsupervised models but could be improved by training in a supervised fashion on interventional spatial transcriptomics data, which might soon be available in high-throughput.

Limitation 2: Our modeling of gene regulation relies on a DAG, and if there are cycles in gene regulation, then the Bayes optimal hypothesis is bound to be outside the set of hypotheses explored by Celcomen. Extending the model to accommodate cyclic regulatory structures perhaps via a causal kinetic model Tejada-Lapuerta et al. (2023) is a promising direction for future work.

Limitation 3: A further limitation of our model is that it does not incorporate cell type information which can be incorporated into both the generative and the inference modules of our Celcomen. For instance, in the inference module, we can infer not only gene-gene matrices, but also gene-gene-CellType-CellType matrices, to allow different gene-gene interactions for each pair of cell types. In the generative module, we could consider cell types when generating the counterfactuals. This amounts to imposing a prior probability for each cell/node in the graph. Incorporating cell type information into Celcomen can also enhance the interpretability of our method, especially in heterogeneous tissues, which have a diverse composition in terms of cell types. On the other hand, incorporating cell type information might create challenges when cell type continua are arbitrarily clustered into separate cell types, and requires the generalization of our Theorem 1 and Theorem 2 to the case of gene-gene-CellType-CellType matrices. Therefore, this interesting and important research direction is left for future work.

Limitation 4: Our modeling of Batch effects as reflected in assumption 3 amounts to the causal sufficiency assumption. Indeed, without it, the identifiability proof would not go through, and the effect of counterfactuals would not be identifiable. We are intrigued by the possibility of weakening the causal sufficiency assumption while preserving the identifiability guarantees of the model and are open to exploring this in future work.

Limitation 5: Our modeling of gene regulation relies only on pairwise gene forces and, as such, omits synergistic effects between genes. A straightforward extension of Celcomen to include such effects could help understand more complex regulatory dynamics.

Limitation 6: Finally, although technically not a limitation, we want to explain that for Celcomen to be able to disentangle intra- and inter-cellular gene regulation, it needs data that are of single-cell resolution. When run on multi-cell resolution data, such as Visium, Celcomen disentangles intra-spot and inter-spot gene regulation. Intra-spot gene regulation would involve signatures of intra-cellular, juxta-crine, and short-range paracrine gene regulation, whereas inter-spot gene regulation would involve long-range paracrine gene regulation.

## K    ENTENDED DATA FIGURES

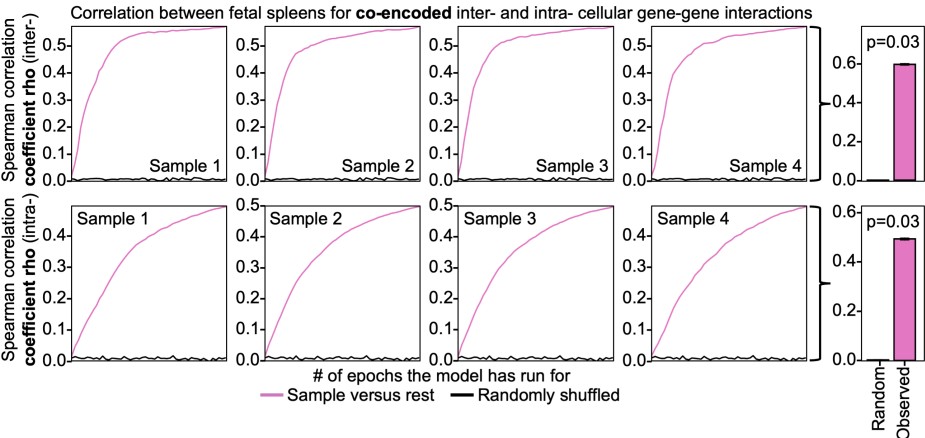

Figure 6: Extended Data Figure 1: Celcomen recapitulates its identifiability guarantees through strong sample-to-sample correlation on real human samples.Left: Line plots with the x-axis as epochs and y-axis as the Spearman correlation coefficients between the gene-gene interaction matrices of the model trained on the specified sample and the model trained on all other samples. The sample utilized for the sample-specific model is annotated directly on the plot. The color of the line, see the lower legend, indicates whether it represents comparisons between the two observed models, pink, or between a random shuffling of the two gene-gene interactions, black, to represent a null model. Right: Bar plots with the left black bar representing the average final Spearman correlation coefficient between randomly shuffled gene-gene interaction matrices of the sample-specific model and model trained on all other samples, and the right pink bar representing the observed correlation. P-values are derived from the Mann-Whitney U test and are annotated directly on the plot. Error bars indicate standard error.

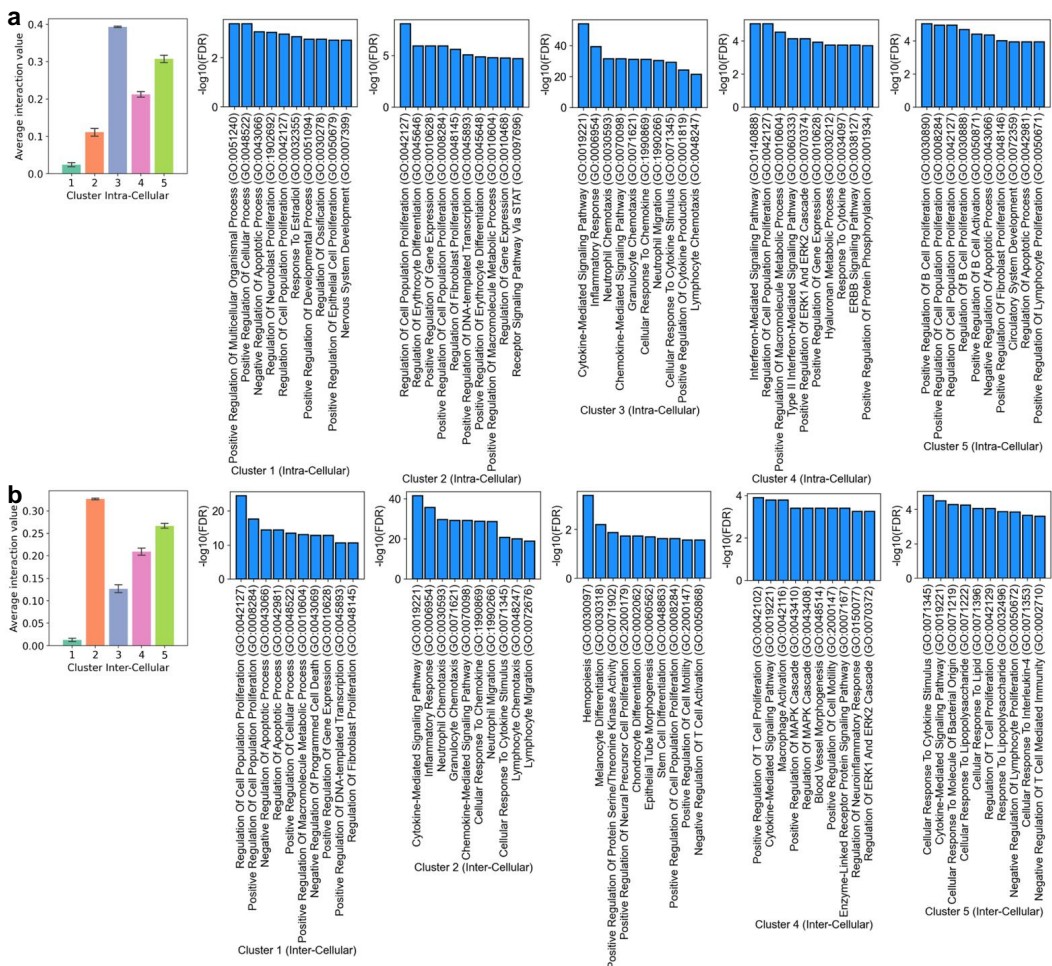

Figure 7: Extended Data Figure 2: Biological interpretation of Gene-Gene Interaction Matrices and Cluster Analysis for Xenium human glioblastoma. (a) Clustered intra-cellular gene-gene interaction matrix, highlighting clusters 3, 4, and 5 as the most strongly interacting intra-cellular modules. Pathway enrichment analysis identifies these clusters as associated with proliferative/apoptosis functions (cluster 5), intracellular signaling (cluster 4), and post-cytokine stimulus response (cluster 3), which are predominantly intra-cellular controlled processes. (b) Clustered inter-cellular gene-gene interaction matrix, emphasizing clusters 2, 4, and 5 as the strongest inter-cellular interactions. Pathway enrichment reveals cytokine response signatures, with inter-cellular cluster 2 capturing markedly more interactions related to cytokine and chemotaxis compared to the analogous intra-cellular cluster. This highlights the model's ability to distinguish inter-cellular interactions selectively.

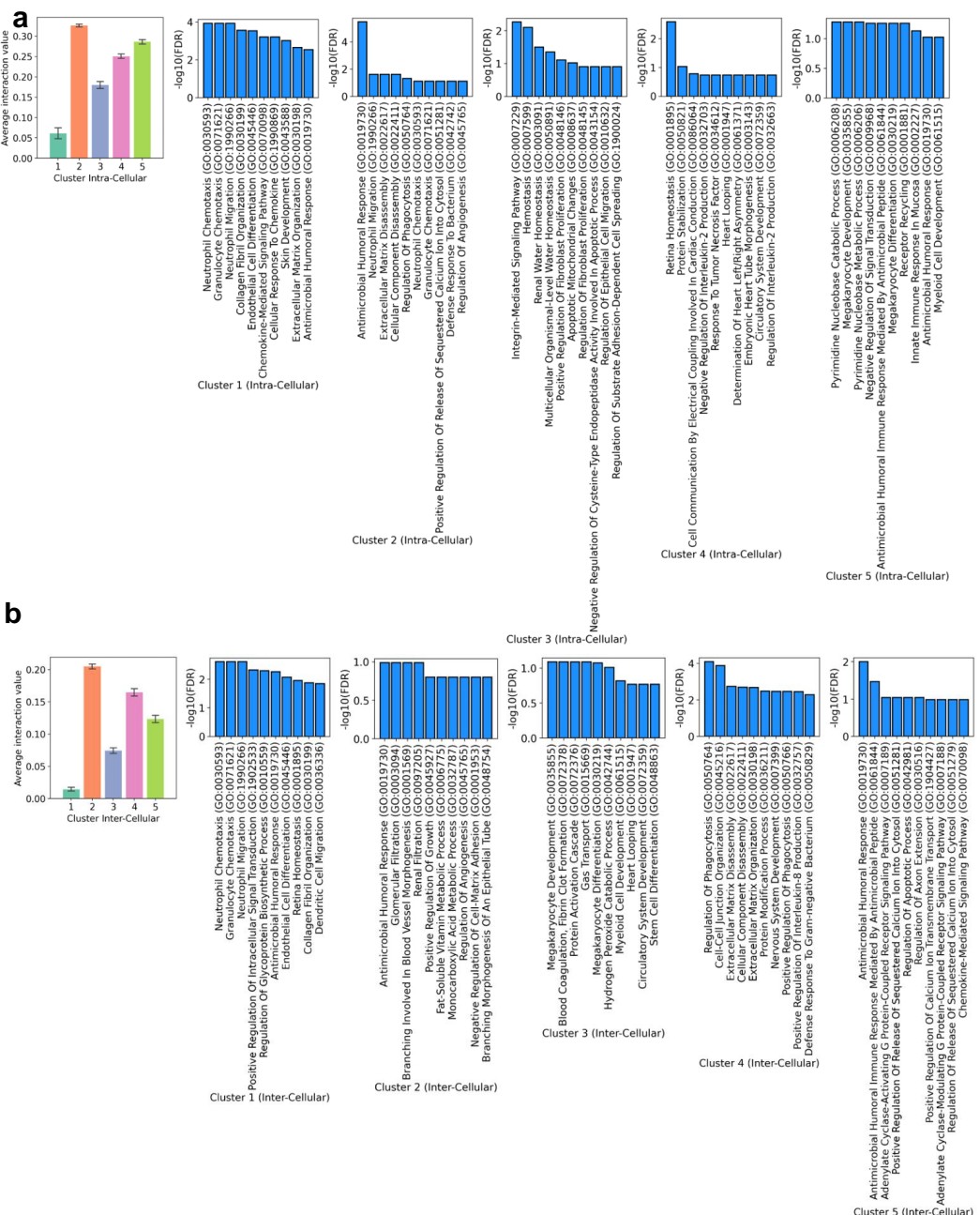

Figure 8: Extended Data Figure 3: Biological interpretation of Gene-Gene Interaction Matrices and Cluster Analysis for Visium human spleen. a) Clustered intra-cellular gene-gene interaction matrix, identifying cluster 2 as having the highest interactions, with cluster 5 as a close second. Both clusters contain genes known to be associated with intracellular processes. Specifically, pathway enrichment analysis reveals that cluster 2 is associated with chemotaxis-related intracellular signaling, including chemokine response pathways and calcium release. Cluster 5 shows enrichment for nucleotide metabolism, potentially linked to the regulation of proliferation via nucleotide availability. b) Clustered inter-cellular gene-gene interaction matrix, highlighting cluster 2 and cluster 4 as the most strongly interacting gene clusters in this matrix. Cluster 2 is enriched for antimicrobial response and angiogenesis programs, while cluster 4 is associated with phagocytosis and extracellular matrix-related functions. All of these tend to be associated with inter-cellular rather than intracellular functions.

