# OpenReview forum: "Estimation of single-cell and tissue perturbation effect in spatial transcriptomics via Spatial Causal Disentanglement"
_ICLR.cc/2025/Conference — ICLR 2025 Poster_

### Official Review · Reviewer_zadU · 2024-10-23

**Soundness:** 3
**Presentation:** 3
**Contribution:** 3
**Rating:** 6
**Confidence:** 3

**Summary:**

In the manuscript, ‘Estimation of single-cell and tissue perturbation effect in spatial transcriptomics via Spatial Causal Disentanglement’, the authors introduce Celcomen, a generative GNN that incorporates causal inference for the task of predicting perturbation effects and counterfactuals, with applications to spatial transcriptomics data and single-cell data. The authors aim, in addition to predictions, to expose mechanistic interpretability, such as the inference of intra-cellular and inter-cellular gene regulation in such data. The method is demonstrated over multiple synthetic and real biological data.

**Strengths:**

- The problem that the authors pose is important and timely
- The method seems to be well grounded and robust
- The demonstrations of the method on real data seems to be promising, especially those having to do with disentangling intra vs inter cellular programs in human glioblastoma spatial transcriptomics data, and showing that in silico knockouts affect signal propagation in a biologically expected manner.

**Weaknesses:**

See the 'Questions' section below for some comments on readability, assumptions made, missing limitations discussion, and presentation of results.

**Questions:**

Comments:
- It’s unclear why the spatial structure of cells in tissues should be captured well by a DAG.
- The text could be improved at multiple points for readability.
- In the related work section, it would be better to add a subsection on generative models and disentanglement for single-cell and spatial transcriptomics data.
 - It would be better to give intuition in the text for the meaning of Theorem 2.
- ‘we observed a strong positive correlation between these two gene-gene interaction matrices’ - there are no results/statistics in the main text, and in the supplement we see a maximum of 0.4/0.5 correlation value, so needs to be rephrased. Also, unclear how these results point to identifiability, as there only seems to be analysis of correlation
- About the actual gene-gene interaction matrices themselves (corresponding to results in Fig. 5) - do these make sense in terms of the current knowledge of existing gene-gene interactions? There are many ways to evaluate these types of questions, and this is crucially missing from the evaluation and results in the paper.
- The discussion section is missing a discussion of the limitations of the approach.

Minor comments:
- some of the references throughout don’t have parentheses around them.
- when introducing spatial transcriptomics, better to give an additional example (or examples), beyond VisiumHD.
- The connections between the visualizations in Fig. 1 and both its caption and the reference to it from the text, are not entirely clear.
- There are some strange phrasing of text throughout. An example (but not the only one): ‘Suppose for instance that nature imposes the colocalization of genes..’
- There are inexact sentences (again, example, but not the only one): “Since half of the time, the nearest neighbor of a nearest neighbor is also a nearest neighbor”
- ‘”Least” here is meant in the sense of entropy’ - needs explanation
- ‘for an ant forced to walk on the surface of a table, this force happens to be electromagnetism’?
- After theorem 1, there are repetitions in the text relative to the numbered equations.
- typo: ‘demonstrates that Celcomen’s usefulness as a toy model for mechanistic interpretability since’, ‘resolutionFarah’

---

> ### Author Response · Authors · 2024-11-19
> **Rebuttal Part 1**
>
> We thank the Reviewer for recognizing the importance and timeliness of the problem we address in our work. We are glad for the positive feedback regarding the robustness and good grounding of Celcomen and are particularly pleased that the Reviewer finds our results on disentanglement and counterfactual prediction to be promising. We also thank the Reviewer for the actionable feedback that helped us to improve our paper. In particular, we have added new experiments to (1) address weakness 6 below; and (2) provide in-vivo validation of our method. We hope that you will find our added experiments and responses satisfactory, and that you will consider revising your score.
>
>
> **Regarding Q1(structure of cells and DAG):**  We thank the Reviewer for bringing up the DAG assumption in causality. In Appendix E “Review of Probably Approximately Correct learning” we discuss PAC learning theory. Our model possesses identifiability guarantees within its space of available hypotheses, which follow a DAG structure. If the Bayes optimal hypothesis for spatial transcriptomics (noted as $h^{**}$ in the paper) does not follow a DAG (e.g., because there exist cells i,j and genes a,b,c,d such that  $s_{ia}$ -> $s_{jb}$ -> $s_{jc}$ -> $s_{id}$ -> $s_{ia}$) then our algorithm will try to find the closest DAG (noted as $h^*$ in the paper)  to approximate the Bayes optimal hypothesis.  As explained in (Tejada-Lapuerta et al, arXiv:2310.14935) a way to avoid the problem of the presence of cycles in gene regulation is to adopt causal kinetic models. Combining the causal kinetic models’ mentality with our models of Virtual Tissues is an intriguing direction to explore in the future.  In our revised paper, we addressed these comments by adding the discussion given here in the Appendix J “Limitations and Future Directions”. Thank you.
>
> **Regarding Q2 (readability)**: Thank you for the comment. We revised our paper accordingly, particularly we added clarifying statements according to reviewers comments in the Motivation section (Section 3.1). All our changes are marked in blue.
>
>
> **Regarding Q3 (additional subsection to related works):** Thank you for the comment. To further distinguish our Celcomen, we added a review of methods relevant for disentanglement and counterfactual predictions in *dissociated* single cell data to the Related Work section (Section 2). Thank you.
>
> **Regarding Q4 (intuition to theorem 2):**  We appreciate your comment. In our paper, we have discussed the intuition behind Theorem 2 in the second paragraph of the Introduction (Section 1) and the paragraph before the statement of Theorem 2. Nonetheless, we agree with the Reviewer that it can be expanded, therefore we expanded Appendix F, which contains the proof of the identifiability theorem, to include more intuition behind it. Specifically, our revisions include the following discussion:
>
> *Identifiability for a model means that a unique configuration of our model fits the observed data. This is very important for causal models since otherwise our model could tell us that A either causes B or it doesn’t cause B, which is an unhelpful tautology.  Identifiability is a point of principle of whether the model would be able to definitively decide that A causes B if provided enough data. For this reason most proofs of identifiability happen in the infinite data limit. For instance, imagine that we want to estimate the causal effect A has on B when there is an unobserved confounder between A and B. In this case, the causal effect could never be identified by any model, regardless of how much data we collect.*
>
> **Regarding Q5 (statistical results):** We thank the reviewer for the comment. We have now mentioned the explicit value of correlation in the main text for clarity. We also added more context to make the connection to identifiability more explicit. Specifically, what we aim to say regarding identifiability is that since our model is identifiable a single unique hypothesis fits the data. If a model is not identifiable, then more than one hypothesis fits the data equally well, and as a result those models randomly pick one of them based on statistical noise. This is why lack of identifiability leads to results that are not robust, since those models might transition to a completely different hypothesis. Since our results remain robust across different samples of the same biological tissue, this is in line with our model being identifiable.

---

> > ### Author Response · Authors · 2024-11-19
> > **Rebuttal Part 2**
> >
> > **Regarding Q6 (gene-gene interaction matrices):** We thank the reviewer for providing us with the opportunity to expand on the biological interpretation of  the gene-gene interaction matrices (inter- and intra- cellular) learned by the Celcomen model. In our original submission, we included evidence that the intra- and inter-cellular gene-gene interaction matrices capture the activity of genes that are cytoplasmic and secreted, respectively (please see Figure 3b). However, we now take the opportunity to provide an orthogonal verification in terms of gene ontologies of the respective gene sets.
> >
> > To understand the learned gene-gene interaction modules, we clustered each interaction matrix into five clusters of genes (n=5 to 20 were searched and n=5 had the best score via the silhouette matrix). To identify which gene clusters were particularly important to a given interaction matrix, we computed the average interaction value of each gene per matrix; clusters where genes have many interactions reported are likely to be important to that interaction module.
> >
> > We here provide the biological interpretation of the gene clusters both for the spleen data experiment in the appendix K and for the human glioblastoma experiment in the main text (Figure 3):
> >
> > **Xenium human glioblastoma (brain cancer), included in the text as extended data figure 2**:
> >
> > For the intra-cellular interaction matrix, we identify intra-cellular clusters 3, 4, and 5 having the greatest interactions captured and thus the model predicts intra-cellular networks to largely be centered around these gene clusters. Upon pathway enrichment analysis, we find these clusters to largely encapsulate proliferative/apoptosis (cluster 5), intracellular signaling (cluster 4), and post-cytokine stimulus response related functions (cluster 3). These are all largely intra-cellular controlled functions.
> >
> > In contrast, for the inter-cellular interaction matrix, we identify inter-cellular clusters 2, 4, and 5 as being most critical to the inter-cellular aspect of the model. Pathway enrichment analysis reveals these clusters as encapsulating cytokine response signatures.
> >
> > Notably, the cytokine and chemotaxis gene cluster in the inter-cellular matrix (inter-cellular cluster 2) has markedly more gene-gene interactions captured than its analogous module in the intra-cellular matrix (intra-cellular cluster 4). This validates the ability for the inter-cellular module to selectively capture inter-cellular interactions (e.g. cytokine response) to a greater degree than the intra-cellular module which we observe above to selectively capture intra-cellular functions (e.g. proliferation and apoptosis).
> >
> > **Visium human spleen, included in the text as extended data figure 2**:
> >
> > For the intra-cellular interaction matrix, we identify cluster 2 as having the greatest interactions captured with cluster 5 a close second. Pathway enrichment analysis points to chemotaxis related intracellular signaling as observed through the chemokine response pathways and release of sequestered calcium which is a known downstream effect of chemokine receptor ligand binding. Cluster 5 presents with less significant enrichment however still points to a clear module of nucleotide metabolism possibly related to the regulation of proliferation through modulation of nucleotide availability.
> >
> > In contrast, in the inter-cellular interaction matrix we find its gene clusters 2 followed by 4 as having captured the greatest number of gene-gene interactions. These interactions encapsulate antimicrobial response and angiogenesis programs, for cluster 2, and phagocytosis and extracellular matrix, for cluster 4, related programs. All of these tend to be associated with inter-cellular rather than intra-cellular functions (i.e. reliant upon secreted and membrane-bound proteins and factors). Thus, analysis of the gene-gene interaction matrices from the trained Celcomen model on all Visium spleen samples provides support for the model to disentangle and separate inter-cellular vs. intra-cellular biological programs.

---

> > > ### Author Response · Authors · 2024-11-19
> > > **Rebuttal Part 3**
> > >
> > > **Regarding Q7 (limitations):** We thank the Reviewer for pointing out the absence of a discussion on the limitations of the approach. We provide here a discussion of the main limitations of our method and have also included it in appendix J of the paper to respect the page limit. We are happy to incorporate it in the main text after acceptance of our paper.
> > >
> > > *Limitation 1:* Unsupervised/Observational learning of causal structure. Celcomen aims to identifiably retrieve the undirected form of the DAG underlying intra- and inter-cellular regulation from observational data, ie without interventional data. Without interventional data it is impossible to uncover more than the Markov equivalence class of the DAG, which is the undirected form of the graph along with partial information about colliders. Therefore Celcomen ability is close to being tight against the best possible bound on the abilities of unsupervised models. Creating a supervised version of Celcomen, meaning one that exploited also interventional data is an exciting direction that is worth future investigation, especially since interventional spatial transcriptomics data might soon be available in high-throughtput.
> > >
> > > *Limitation 2:* Our modeling relies on a DAG and if biology has cycle in its gene regulation then the Bayes optimal hypothesis is bound to be outside the set of hypotheses explored by Celcomen. We could pursue a causal kinetic modeling version of Celcomen in future work to include hypotheses with cycles in gene regulation, as we mention above in weakness 1.
> > >
> > > *Limitation 3:* A further limitation of our model is that it does not incorporate cell type information and, therefore, cannot learn cell type pair-specific gene-gene interactions.
> > >
> > > *Limitation 4:* Our modeling of Batch effects, as reflected in assumption 3, amounts to the causal sufficiency assumption. Indeed without it the identifiability proof wouldn’t go through and the effect of counterfactuals would not be identifiable. We are intrigued by the possibility of weakening the causal sufficiency assumption while preserving the identifiability guarantees of the model, and are open to exploring this in future work.
> > >
> > > *Limitation 5:* Our modeling of gene regulation relies only on pairwise gene forces, and as such omits synergistic effects between genes which can be helpful in modeling more detailed aspects of complex cases of gene regulation.
> > >
> > >
> > > **Regarding minor M1 (format of references):** We thank the reviewer for pointing out that some references did not have parentheses around them. It has now been fixed in the revised paper.
> > >
> > > **Regarding minor M2 (examples of transcriptomics):**  We thank the reviewer for this comment. We have now additionally cited Xenium as well as the original Curio and Stereo-seq, as such we now mention both imaging and sequencing based experimental platforms from different companies that profile spatial transcriptomics with single-cell resolution.
> > >
> > > **Regarding minors M3-M7 (editing of text):** We sincerely appreciate your comments. We have worked to address and implement them in our revised paper. Thank you for the important suggestions.
> > >
> > > **References**
> > >
> > > Tejada-Lapuerta, Alejandro, et al. "Causal machine learning for single-cell genomics." arXiv preprint arXiv:2310.14935 (2023)
> > >
> > > Maxime Dhainaut, Samuel A Rose, Guray Akturk, Aleksandra Wroblewska, Sebastian R Nielsen, Eun Sook Park, Mark Buckup, Vladimir Roudko, Luisanna Pia, Robert Sweeney, et al. Spatial crispr genomics identifies regulators of the tumor microenvironment. Cell, 185(7):1223–1239, 2022.

---

> > > > ### Comment · Reviewer_zadU · 2024-11-27
> > > >
> > > > I appreciate the efforts done by the authors to thoroughly address all the comments and suggestions raised during the review process.

---

### Official Review · Reviewer_pnJL · 2024-11-03

**Soundness:** 3
**Presentation:** 3
**Contribution:** 3
**Rating:** 8
**Confidence:** 3

**Summary:**

The authors present a novel method for learning causal relationships in the context of spatial transcriptomics data, namely Celcomen. The method is devised to distinguish between inter and intra cellular causal interactions between genes. The authors characterize their method both from a theoretical point of view, demonstrating Celcomen identifiability (granted that assumptions hold), as well as from a practical point of view, applying their methods on real-world data.

**Strengths:**

The work has very strong theoretical basis. The authors provide proof of the identifiability of their method; moreover, they also honed their implementation so that to tackle large scale problem, for example by introducing a new Mean Field Theory approximation.

**Weaknesses:**

The main weakness of the manuscript is its poor connection to the underlying biology, as exemplified by the two issues below.

- The authors present a set of three assumptions upon which their whole model relies. However, they never discuss the implications of these assumptions, and without appropriate commentary, I frankly find difficult to understand what these assumptions entails. How do their assumptions compare with respect to the current limits of spatial transcriptomics technologies? Are there specific conditions, tissues, or other situations where these assumptions may not hold? Moreover, how are these assumptions related to more classical assumptions of causal learning theory, namely faithfulness, Markov condition, and causal sufficiency?

- Cells in tissues are usually grouped within functional clusters. Most typically, cells will belong to distinct cell types, which depending on the type of tissue may located together (endothelial cells in the inner layers of blood vessels) or scattered across the connecting matrix (e.g., fibroblast in mussle tissue). Other cases are possible as well: for example, glutamatergic neurons in the cortex will belong to different layers, where each layer will have different roles in cognitive functions. To my understanding, in its current iteration the Celcomen method fully disregards information on cell types or other functional clustering, even if this information can be instrumental in better understanding cell to cell communication mechanisms. For example, causal relationships between cell may be largely dictate on whether these cells are of the same or of different type.

**Questions:**

I would invite the authors to add an extensive paragraph regarding their assumptions:
- Discuss the biological and technological implications of each assumption
- Compare the assumptions to current limitations of spatial transcriptomics technologies
- Identify specific conditions or tissues where the assumptions may not hold
- Relate the assumptions to classical causal learning assumptions like faithfulness, Markov condition, and causal sufficiency

Modifying Celcomen so that to include cell type information is an onerous task that might deserve a separate publication. However, I would suggest that the authors:
- Acknowledge this limitation of their current model
- Discuss how disregarding cell type information may impact the interpretation of their results
- Propose specific ways the model could be extended in future work to incorporate cell type or functional cluster information

---

> ### Author Response · Authors · 2024-11-19
> **Rebuttal Part 1**
>
> We thank the Reviewer for the kind remarks about the strong theoretical underpinnings of Celcomen, such as the identifiability Theorem, and the Lemmas deriving a new mean field theory approximation that allows to speed up inference. We are also grateful for the constructive feedback regarding connections to the underlying biology, which we address one by one in our rebuttal and new experiments below. We hope that in light of our responses and revision of the paper, you will consider revising your score.
>
> **Regarding W1 (assumptions):** We agree with the Reviewer that a bigger discussion of biology would help the reader. To address your comment, we now elaborate on our assumptions. Specifically,  we would like to point out that our assumptions are motivated by biological concepts, as follows:
>
> *Assumption 1  (nearest neighbor assumption):* Cell-cell communication and ligand-receptor interactions are often reflected in local tissue architecture (Ren et al, Cell Research 2020), which mathematically translates into our nearest neighbor assumption. Several methods are based on similar assumptions that a cell’s ligand affect or at least are correlated with their neighbors’ expression of receptor genes. For instance in (Birk et al bioArxiv 2024) they include a term that tries to regress the value of the receptor gene expression in a cell based on the expression of ligand gene in its neighbors.
>
> *Assumption 2 (Batch effect assumption, also known as causal sufficiency):* By avoiding explicit batch correction, we require that there are no unobserved confounders (often referred to by biologists as batch effects) such as differences stemming from different experimental study sites, different ages or disease states of donors etc.
>
> Additionally, we note that, without interventional data, it is mathematically impossible (without extra assumptions) to infer anything more than the Markov equivalence of the underlying DAG, which is the undirected graph plus information about colliders. Indeed our method obeys this mathematical bound, and Theorem 2 guarantees the identifiability of the undirected DAG.
>
> The assumptions were added to our revised paper in Appendix J “Limitations and future directions”. Thank you.
>
> **Regarding W2 (cells in tissues):** We thank the reviewer for their comments and the exciting prospect of including cell type information. We think that the raised idea can be incorporated both in the generative and the inference modules of our Celcomen. For instance, in the inference module, we can infer not just gene-gene matrices but gene-gene-CellType-CellType matrices so that we can allow different gene-gene interactions for each pair of cell types. In the generative module, we could take cell types into account when generating the counterfactuals. This amounts to imposing a prior probability for each cell/node in the graph.
>
> On the other hand, incorporating cell type information might also create problems when cell type continua are arbitrarily clustered into separate cell types, which is why it merits a separate future study as an important and difficult task. Moreover, the 4D tensor required for gene-gene-CellType-CellType forces might not require modifications to Theorems 1 and 2, which form the main novelty and motivation of the paper. All those directions are indeed exciting, and we will pursue them in future work. We added part of this discussion in the Appendix J “Limitations and future directions”. Thank you.

---

> > ### Author Response · Authors · 2024-11-19
> > **Rebuttal Part 2**
> >
> > **Regarding Q1-4 (biological and causal implications of assumptions):** We thank the Reviewer for the questions and suggestions that helped us to improve the paper. We answer questions 1-4 here, and also kindly refer the Reviewer to our answers from W1 and W2, that are relevant to the questions.  Our responses are as follows:
> >
> > 1. We added clarifying remarks about the implications of our assumptions and their connections to **terminology in causal inference literature** in the “Model Assumptions” Section (section 3.2). Moreover, we included a thorough discussion of Celcomen’s limitations that arise from those assumptions in the Appendix J “Limitations and future directions”.  In the latter Section, we also address the potential extension to include **cell type** information in future work which would help enhance the biological interpretability of our method in heterogeneous tissues (for more details on the limitations due to cell type, please refer to our answer to Q5 below).
> >
> > 2. In regards to how the studied **tissue** relates to our assumptions, we again want to emphasize that in heterogeneous tissues the interpretability of our results could be enhanced by adding this information to our model. This however is left as a direction to explore in the future.
> >
> > 3. With respect to how **existing technology** relates to our assumptions, we note that in order for Celcomen to disentangle intra- and inter-cellular gene regulation, the data needs to be of single-cell resolution. If applied to spatial data whose spots contain many cells, then Celcomen disentangles intra- and inter-**spot** gene regulation. Since there are now several spatial platforms with single-cell (or even sub-cellular) resolution, such as VisiumHD, Xenium, Curio, Stereo-seq, we believe that Celcomen will assist researchers in disentangling inter- and intra-cellular gene regulation. We added a discussion of this in the Appendix J “Limitations and future directions”.
> >
> >
> > **Regarding Q5 (limitations and future works for cell type):** We thank the reviewer for their comments and the exciting prospect of including cell type information in future works. In terms of limitations and future works, we now added “limitations and future directions” in Appendix J. In particular, our response includes: possibility of combining Celcomen with cell type information, kinematic modeling, interventional spatial data (since such datasets are likely to be generated in the near future),  and synergistic effects between genes to allow modeling even more complex gene regulation.
> >
> > **References**
> >
> > Ren, X., Zhong, G., Zhang, Q. et al. Reconstruction of cell spatial organization from single-cell RNA sequencing data based on ligand-receptor mediated self-assembly. Cell Res 30, 763–778 (2020). https://doi.org/10.1038/s41422-020-0353-2
> >
> > Maxime Dhainaut, Samuel A Rose, Guray Akturk, Aleksandra Wroblewska, Sebastian R Nielsen, Eun Sook Park, Mark Buckup, Vladimir Roudko, Luisanna Pia, Robert Sweeney, et al. Spatial crispr genomics identifies regulators of the tumor microenvironment. Cell, 185(7):1223–1239, 2022.
> >
> > Sebastian Birk, Irene Bonafonte-Pard`as, Adib Miraki Feriz, Adam Boxall, Eneritz Agirre, Fani Memi, Anna Maguza, Rong Fan, Gonc¸alo Castelo-Branco, Fabian J. Theis, Omer Ali Bayraktar, Carlos Talavera-L´opez, and Mohammad Lotfollahi. Quantitative characterization of cell niches in spatial atlases. bioRxiv, 2024. doi: 10.1101/2024.02.21.581428. URL https://www.biorxiv.org/content/early/2024/03/21/2024.02.21.581428.

---

> > > ### Comment · Reviewer_pnJL · 2024-11-27
> > >
> > > I commend the authors for the notable effort spent in reviewing their submission. All my comments were answered satisfactorily, and I also appreciate the way the authors replied to the comments from the other reviewers. The presentation of the text has improved, and, consequently, the scope, relevance, applicability and limitations of the proposed method are much clearer.
> > > Therefore, I am modifying my score, to reflect the new status of the paper.
> > >
> > > My only comment at this stage: it is not clear to me why the authors did not include in Appendix J all the interesting discussion points on cell type information that they have formulated above. Right now, they only mention the cell type issue quite rapidly in "Limitation number 3". I would suggest to expand this topic in Appendix J. However, I leave this to the discretion of the authors.

---

> > > > ### Author Response · Authors · 2024-11-28
> > > >
> > > > Dear Reviewer pnJL,
> > > >
> > > > We thank you for recognizing our efforts, and for raising your score. \
> > > > We agree with the Reviewer regarding Appendix J, and following your suggestion, we have now updated our paper to further elaborate on how including cell type could enhance the interpretability and the expressiveness of our model in Appendix J.
> > > >
> > > > Thank you for the valuable feedback that helped us to improve the quality of our paper.
> > > >
> > > > Best regards,
> > > >
> > > > The Authors.

---

### Official Review · Reviewer_KiqG · 2024-11-05

**Soundness:** 3
**Presentation:** 3
**Contribution:** 4
**Rating:** 6
**Confidence:** 3

**Summary:**

This paper introduces Celcoman, a physics inspired generative graph neural network to infer intra- and inter-cellular gene regulation in spatial transriptomics and single-cell data. Celcoman can also be used to simulate the perturbation effect (such as gene knock off) in silico, offering insights into experimentally inaccessible states.

This experiment is validated on both synthetic and real datasets, which includes both glioblastoma and fetal spleen samples.

**Strengths:**

1. I agree with the author's claim that this is the first causally identifiable model for spatial transcriptomics data.
2. The motivation is well explained and the results on the impact of perturbed spot on surrounding cells are very convincing. It shows why we need to consider cell communication on spacial omnics data.
3. The idea is novel. The author links GRN inference among neighboring cells with entropy maximization under the control of some lagrangian multipliers, which are then modeled by GNN. Theorem 1 is also inspiring.

**Weaknesses:**

1. Notation issue. The notations in this paper are not consistent. For example, the author used $s^i_{\alpha}$, $s^\alpha_{i}$, and $s_{\alpha i}$ to refer the same concept.

2. While I understand this is one of the first few papers studying, the lack of baseline comparisons in the experiments makes the results section less convincing. For example, while the results does show the impact of perturbed spot on surrounding area, how accurate it is? With GNN, I'm not surprised to see this kind of phenomenon. Are there any ways to provide a quantitative measurement or comparisons with the methods? Could you add some baselines or ground truth comparison?

3. A common acknowledged fact is that for causal models, there is a huge gap between synthetic data and the data we observe in real life. While the idea of in-silico perturbation data generation is very interesting, can you discuss the similarity/difference between the generated data and real data?

**Questions:**

1. In section 3.1, you mentioned that the inspiration of this work comes from the notion of force in physics and the objective is to learn the "least" number of forces. Then you said "'least' here is in the sense of entropy". However, later on, all we are trying to do is to maximize entropy. After thinking, I think I sort of understand the concept but someone else may get stuck too. Could you add a bit more explanation on how does the objective of minimizing number of forces transform into maximizing the entropy?

2. Could you comment on the runtime of this algorithm? Also, how long does it take (clock time) to finish the experiment you did in your study?

---

> ### Author Response · Authors · 2024-11-19
> **Rebuttal Part 1**
>
> We thank the Reviewer for the positive review and the kind words. We agree with the Reviewer,  that Celcomen, to the best of our knowledge, is novel in terms of an identifiable method for spatial transcriptomics analysis, and Celcomen can be valuable to the community. We also agree that an important feature of Celcomen is that its architecture is mathematically derived from biologically motivated assumptions according to Theorem  1.
>
> We are also thankful for the constructive feedback that helped us improve the paper by adding one new in-vivo experiment and refining the clarity of the text. Below we provide our responses to each of your comments. We hope that you find them satisfactory, and that you will consider revising your score.
>
> **Regarding W1 (notation):**  We thank the reviewer for pointing out these typos. We addressed them in the text to increase clarity. For example, $s_{ia}$ is now corrected to $s_i^a$.
>
> **Regarding W2 (Baselines and Ground truth):**
> We agree with the Reviewer that there is a lack of what we consider models of Virtual Tissues to compare against. There are models of Virtual Cells like Biolord (Piran et al, Nat Biotech 2024) and GEARS (Roohani et at, Nat Biotech  2024) that predict the response of a cell given a perturbation of its environment or its genes. What we aim to do here is essentially the converse: determine not only how the cell is affected by its environment, but also how it *affects* its environment. Moreover, most methods of virtual cells are supervised, whereas our virtual model of tissues is unsupervised allowing for unbiased discovery not limited by current knowledge.  We added this important distinction in the introduction to the paper.
>
> *Furthermore*, we agree with the reviewer that although the results of the perturbation seem sensible, 1) more quantitative assessment against ground truth real data and 2) comparison to baselines would consolidate our arguments. To directly provide a benchmarking of our model to ground truth experimental data, we validate Celcomen on *in vivo* data, see new Figure 4 in the main text. Moreover to directly address the reviewer’s question about more comparisons to baselines we *compare our method to a random baseline*, see new Figure 4 in the main text. To our knowledge, the only dataset that has measured gene knockouts via spatial transcriptomics is Perturb-map (Dhainaut et al, Cell 2022) which leverages the non-HD version of the 10x Genomics Visium technology; notably, this is not single-cell resolution because each spot contains around seven cells on average and thus represents cell communities. Thus, our model is disentangling intra- and inter-spot gene regulation, namely intracellular+juxtacrine+paracrine (short range) regulation from paracrine (long range) regulation, rather than intra- and inter-cellular gene regulation. When we challenged our model with this ground truth we observed very promising results with in silico and in vivo perturbed transcriptomes appearing well correlated and well beyond what a random baseline could achieve. The discussions above were added to the revised paper section 4.3. Thank you.
>
> **Regarding W3 (real-life experiment):** Thank you for the inspiring suggestion. The often large difference between synthetic and real data is a relevant point raised by the Reviewer. Even more to the reviewer’s point, there are also large differences between in-vitro experiments (performed in controlled laboratory settings, such as organoids, petri dishes or test tubes) and in-vivo experiments which are conducted in the context of a full living organism. To address your comment, we validate our predictions on real data using the Perturb-Map public dataset (Dhainaut et al, Cell 2022) of whole transcriptome profiling for in-vivo spatial gene knock-outs. Moreover, we benchmark our performance against a random baseline.
> As described above, across all five lesions, we observe a 0.3-0.5 Spearman correlation between our in-silico perturbation changes and the measured in-vivo perturbation changes, which is far above what can be explained by random baseline (p=0.0079). In summary, in our simulation, our real data experiment and our in-vivo experiment we confirm celcomen’s ability to predict spatial counterfactuals. The results on the new in-vivo experiment are presented in figure 4 and discussed in section 4.3.

---

> > ### Author Response · Authors · 2024-11-19
> > **Rebuttal Part 2**
> >
> > **Regarding Q1 (least entropy):** We thank the Reviewer for the thorough reading of our paper and for the important question. In the paper, we use the word force in the same sense as cause. What we mean to say is that not all correlations observed need to have a cause/force behind them (to minimize the entropy). If A causes B and B causes C, A will likely have a correlation with A, although, by construction, it does not cause C. *An identifiable model* would be able to know that in this situation, we need just 2 forces (not 3) to explain all 3 non-zero pairwise correlations. Hence, in this case, the minimum number of forces required is 2. We added clarifying remarks for this important point in section Motivation (section 3.1). Thank you.
> >
> > **Regarding Q2 (runtimes):** We thank the Reviewer for the question, which allows us to expand on design aspects for Celcomen.
> > Our derivation of the mean field approximation allows us to speed up the computation of Z, the partition function. In particular, with Equation (26) which is the un-approximated formula that derives from theorem 1, computing  Z would require summing all possible spatial transcriptomics samples, which we estimate to be roughly equal to $10^{6,000,000}$ possibilities. Performing this calculation in every training step is computationally infeasible. *On the other hand*, using Equation (49), which describes our Celcomen, Z can be calculated by summing over $10^6$ gene-pairs. Using this approach, Celcomen’s *inference* module runs in 11 seconds for ~500 genes and ~2500 spots.
> >
> > In early phases of the design of the model,  we experimented with sampling from a space of $10^{6,000,000}$ possibilities using MCMC or Gibbs sampling, which proved to be computationally intractable. That is the reason we implemented the generative module of Celomen to generate the most likely spatial transcriptomics sample based on its learned distribution of observed samples. Thanks to this, running the *generative* module for ~500 genes and ~2500 spots takes a few hours on CPU or 12 minutes on a GPU. We added the runtimes and discussions to the revised paper in Appendix I “Computational Efficiency”. Thank you.
> >
> > **References**
> >
> > Zoe Piran, Niv Cohen, Yedid Hoshen, and Mor Nitzan. Disentanglement of single-cell data with biolord. Nature Biotechnology, pp. 1–6, 2024.
> >
> > Yusuf Roohani, Kexin Huang, and Jure Leskovec. Predicting transcriptional outcomes of novel multigene perturbations with gears. Nature Biotechnology, 42(6):927–935, 2024.
> >
> > Maxime Dhainaut, Samuel A Rose, Guray Akturk, Aleksandra Wroblewska, Sebastian R Nielsen, Eun Sook Park, Mark Buckup, Vladimir Roudko, Luisanna Pia, Robert Sweeney, et al. Spatial crispr genomics identifies regulators of the tumor microenvironment. Cell, 185(7):1223–1239, 2022.

---

### Author Response · Authors · 2024-11-19

# Authors' Response Summary

We would like to express our gratitude to all reviewers for their valuable feedback.
Overall, the reviewers appreciated:
The challenging problem we aim to solve, and the novelty of our approach. Reviewer **KiqG** recognizes that our Celcomen **“is the first causally identifiable model for spatial transcriptomics data”**, **“the idea is novel,” and that the “motivation is well explained”**. Reviewer **pnJL** acknowledges that we “**present a novel method** for learning causal relationships”.   Reviewer **zadU** highlights that **“The problem that the authors pose is important and timely”**.

The mathematical foundation of our approach. Reviewer **KiqG** said, regarding the derivation of Celcomen’s architecture, that **“Theorem 1 is inspiring”**; reviewer **pnJL** said that Celcomen **“has very strong theoretical basis”** in terms of its identifiability proof and our derivation of formulas to scale our approach to large graphs; and Reviewer **zadU** stated that **Celcomen “seems to be well grounded and robust”**.

The comprehensive study of disentanglement, identification of causal links, and counterfactual predictions on real and synthetic data. Reviewer **KiqG** highlights that the **“results on the impact of perturbed spot on surrounding cells are very convincing”** and Reviewer **zadU** comments that **“demonstrations of the method on real data seems to be promising”.**
Your thoughtful comments and suggestions allowed us to improve our paper, and we provided individual responses to each reviewer. We hope that you will find them satisfactory, and that you will consider revising your score. We are happy to discuss existing or additional questions and suggestions you may have.  Below, we provide a summary of the changes made to our paper during the rebuttal period.



**New Experiments.** Several additional experiments were conducted based on the reviewers’ comments, as follows:

1) In-vivo validation of Celcomen’s spatial gene-knockout predictions (section 4.3 and figure 4) on the Perturb-map (Dhainaut et al, Cell 2022) dataset of full-transcriptome profiling for spatial gene-knockouts. To the best of our knowledge,  it is the only currently published dataset of this kind.  Moreover, to our knowledge Celcomen is the first learning model (of  Virtual Tissues) to be applied on in-vivo data for spatial gene-knockouts. Further consolidating the novelty of the work presented in this paper.

2) Comparison of Celcomen to random baselines in the in-vivo data (section 4.3 and figure 4). Similarly, to our knowledge, this is the first attempt to benchmark learning methods on the task of tissue response to spatial gene-knockouts.

3) Biological investigation of the intra and inter-cellular gene-gene force matrices in the spleen Visium data (Extended data figure 2) and in the brain Xenium data (Extended data figure 3). These investigations provide further evidence that Celcomen’s predictions are biologically sensible even when there is no experimental or synthetic ground truth.


**Revisions to the paper.**  Some of the existing sections of the paper were revised to increase clarity based on the reviewers’ comments, and new sections were added following your guidance:

1. We added section 4.3 for the in-vivo experiments.
2. We edited the motivation and inspiration section (section 3.1) to increase clarity and  address comments by reviewer zadU.
3. We changed the Assumptions section (section 3.2) to relate it to terminology from the causal inference literature.
4. We added an extensive discussion of the limitations of our assumptions and potential future directions of our method as Appendix J.
5. We added two more extended data figures in Appendix K to offer even more biological interpretation of the intra and inter-cellular gene-gene force matrices in the spleen Visium and the brain Xenium data.


---
All the revisions and additions to the paper are **marked in blue**, for your convenience. We hope that you will find the new experiments, responses and edits satisfactory, and that you will consider revising your score.

---

### Comment · Area_Chair_SwLU · 2024-11-26

Dear Reviewers KiqG, pnJL, zadU,
If not already, could you please take a look at the authors' rebuttal? Thank you for this important service.
-AC

---

### Meta-Review · Area_Chair_SwLU · 2024-12-19

**Metareview:**

This paper proposes a generative graph neural network to infer and disentangle the causal structure diagram of feature interactions from spatial samples such as spatial transcriptomics data. The method can also be used to predict spatial perturbation effect in silico. Empirical successes are demonstrated on both synthetic and real-life biological data. Some of the reviewers (e.g., pnJL) also found the theoretical component to be strong. Overall, both reviewers and I agree that this work makes both biological and machine learning contributions that warrant its acceptance.

**Additional Comments On Reviewer Discussion:**

Most of the reviewers' concerns have been addressed during the rebuttal.

---

### Decision · Program_Chairs · 2025-01-22

Accept (Poster)